# FEATURE-ROBUST OPTIMAL TRANSPORT
# FOR HIGH-DIMENSIONAL DATA

## ABSTRACT

Optimal transport is a machine learning problem with applications including distribution comparison, feature selection, and generative adversarial networks. In this paper, we propose feature-robust optimal transport (FROT) for high-dimensional data, which solves high-dimensional OT problems using feature selection to avoid the curse of dimensionality. Specifically, we find a transport plan with discriminative features. To this end, we formulate the FROT problem as a min–max optimization problem. We then propose a convex formulation of the FROT problem and solve it using a Frank–Wolfe-based optimization algorithm, whereby the subproblem can be efficiently solved using the Sinkhorn algorithm. Since FROT finds the transport plan from selected features, it is robust to noise features. To show the effectiveness of FROT, we propose using the FROT algorithm for the layer selection problem in deep neural networks for semantic correspondence. By conducting synthetic and benchmark experiments, we demonstrate that the proposed method can find a strong correspondence by determining important layers. We show that the FROT algorithm achieves state-of-the-art performance in real-world semantic correspondence datasets.

## 1 INTRODUCTION

Optimal transport (OT) is a machine learning problem with several applications in the computer vision and natural language processing communities. The applications include Wasserstein distance estimation (Peyré et al., 2019), domain adaptation (Yan et al., 2018), multitask learning (Janati et al., 2019), barycenter estimation (Cuturi & Doucet, 2014), semantic correspondence (Liu et al., 2020), feature matching (Sarlin et al., 2019), and photo album summarization (Liu et al., 2019). The OT problem is extensively studied in the computer vision community as the earth mover's distance (EMD) (Rubner et al., 2000). However, the computational cost of EMD is cubic and highly expensive. Recently, the entropic regularized EMD problem was proposed; this problem can be solved using the Sinkhorn algorithm with a quadratic cost (Cuturi, 2013). Owing to the development of the Sinkhorn algorithm, researchers have replaced the EMD computation with its regularized counterparts. However, the optimal transport problem for high-dimensional data has remained unsolved for many years.

Recently, a robust variant of the OT was proposed for high-dimensional OT problems and used for divergence estimation (Paty & Cuturi, 2019; 2020). In the robust OT framework, the transport plan is computed with the discriminative subspace of the two data matrices $X \in \mathbb{R}^{d \times n}$ and $Y \in \mathbb{R}^{d \times m}$. The subspace can be obtained using dimensionality reduction. An advantage of the subspace robust approach is that it does not require prior information about the subspace. However, given prior information such as feature groups, we can consider a computationally efficient formulation. The computation of the subspace can be expensive if the dimensionality of data is high, for example, $10^4$.

One of the most common prior information items is a feature group. The use of group features is popular in feature selection problems in the biomedical domain and has been extensively studied in Group Lasso (Yuan & Lin, 2006). The key idea of Group Lasso is to prespecify the group variables and select the set of group variables using the group norm (also known as the sum of $\ell_2$ norms). For example, if we use a pretrained neural network as a feature extractor and compute OT using the features, then we require careful selection of important layers to compute OT. Specifically, each

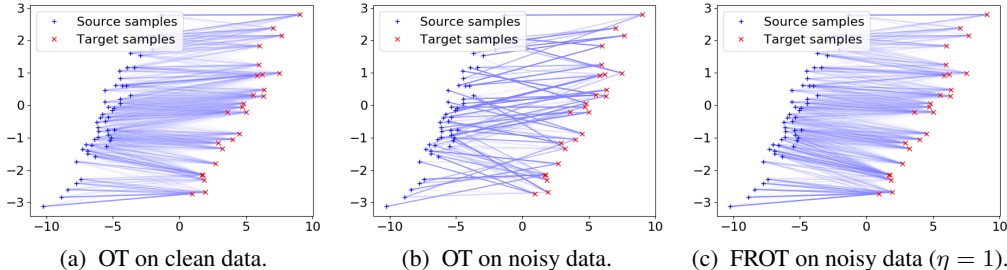

| (a) OT on clean data. | (b) OT on noisy data. | (c) FROT on noisy data ($\eta = 1$). |

Figure 1: transport plans between two synthetic distributions with 10-dimensional vectors $\widetilde{\boldsymbol{x}} = (\boldsymbol{x}^\top, \boldsymbol{z}_x^\top)$, $\widetilde{\boldsymbol{y}} = (\boldsymbol{y}^\top, \boldsymbol{z}_y^\top)$, where two-dimensional vectors $\boldsymbol{x} \sim N(\boldsymbol{\mu}_x, \boldsymbol{\Sigma}_x)$ and $\boldsymbol{y} \sim N(\boldsymbol{\mu}_y, \boldsymbol{\Sigma}_y)$ are true features; and $\boldsymbol{z}_x \sim N(\boldsymbol{0}_8, \boldsymbol{I}_8)$ and $\boldsymbol{z}_y \sim N(\boldsymbol{0}_8, \boldsymbol{I}_8)$ are noisy features. (a) OT between distribution $\boldsymbol{x}$ and $\boldsymbol{y}$ is a reference. (b) OT between distribution $\widetilde{\boldsymbol{x}}$ and $\widetilde{\boldsymbol{y}}$. (c) FROT transport plan between distribution $\widetilde{\boldsymbol{x}}$ and $\widetilde{\boldsymbol{y}}$ where true features and noisy features are grouped, respectively.

layer output is regarded as a grouped input. Therefore, using a feature group as prior information is a natural setup and is important for considering OT for deep neural networks (DNNs).

In this paper, we propose a high-dimensional optimal transport method by utilizing prior information in the form of grouped features. Specifically, we propose a feature-robust optimal transport (FROT) problem, for which we select distinct group feature sets to estimate a transport plan instead of determining its distinct subsets, as proposed in (Paty & Cuturi, 2019; 2020). To this end, we formulate the FROT problem as a min–max optimization problem and transform it into a convex optimization problem, which can be accurately solved using the Frank–Wolfe algorithm (Frank & Wolfe, 1956; Jaggi, 2013). The FROT's subproblem can be efficiently solved using the Sinkhorn algorithm (Cuturi, 2013). An advantage of FROT is that it can yield a transport plan from high-dimensional data using feature selection, using which the significance of the features is obtained without any additional cost. Therefore, the FROT formulation is highly suited for high-dimensional OT problems. Through synthetic experiments, we initially demonstrate that the proposed FROT is robust to noise dimensions (See Figure 1). Furthermore, we apply FROT to a semantic correspondence problem (Liu et al., 2020) and show that the proposed algorithm achieves SOTA performance.

**Contribution:**

- We propose a feature robust optimal transport (FROT) problem and derive a simple and efficient Frank–Wolfe based algorithm. Furthermore, we propose a feature-robust Wasserstein distance (FRWD).

- We apply FROT to a high-dimensional feature selection problem and show that FROT is consistent with the Wasserstein distance-based feature selection algorithm with less computational cost than the original algorithm.

- We used FROT for the layer selection problem in a semantic correspondence problem and showed that the proposed algorithm outperforms existing baseline algorithms.

## 2 BACKGROUND

In this section, we briefly introduce the OT problem.

**Optimal transport (OT):** The following are given: independent and identically distributed (i.i.d.) samples $\boldsymbol{X} = \{\boldsymbol{x}_i\}_{i=1}^n \in \mathbb{R}^{d \times n}$ from a $d$-dimensional distribution $p$, and i.i.d. samples $\boldsymbol{Y} = \{\boldsymbol{y}_j\}_{j=1}^m \in \mathbb{R}^{d \times m}$ from the $d$-dimensional distribution $q$. In the Kantorovich relaxation of OT, admissible couplings are defined by the set of the transport plan:

$$\boldsymbol{U}(\mu, \nu) = \{\boldsymbol{\Pi} \in \mathbb{R}_+^{n \times m} : \boldsymbol{\Pi} \boldsymbol{1}_m = \boldsymbol{a}, \boldsymbol{\Pi}^\top \boldsymbol{1}_n = \boldsymbol{b}\},$$

where $\boldsymbol{\Pi} \in \mathbb{R}_+^{n \times m}$ is called the transport plan, $\boldsymbol{1}_n$ is the $n$-dimensional vector whose elements are ones, and $\boldsymbol{a} = (a_1, a_2, \ldots, a_n)^\top \in \mathbb{R}_+^n$ and $\boldsymbol{b} = (b_1, b_2, \ldots, b_m)^\top \in \mathbb{R}_+^m$ are the weights. The OT problem between two discrete measures $\mu = \sum_{i=1}^n a_i \delta_{\boldsymbol{x}_i}$ and $\nu = \sum_{j=1}^m b_j \delta_{\boldsymbol{y}_j}$ determines the

optimal transport plan of the following problem:

$$\min_{\mathbf{\Pi} \in \boldsymbol{U}(\mu,\nu)} \quad \sum_{i=1}^{n} \sum_{j=1}^{m} \pi_{ij} c(\boldsymbol{x}_i, \boldsymbol{y}_j), \tag{1}$$

where $c(\boldsymbol{x}, \boldsymbol{y})$ is a cost function. For example, the squared Euclidean distance is used, that is, $c(\boldsymbol{x}, \boldsymbol{y}) = \|\boldsymbol{x} - \boldsymbol{y}\|_2^2$. To solve the OT problem, Eq. (1) (also known as the earth mover's distance) using linear programming requires $O(n^3), (n = m)$ computation, which is computationally expensive. To address this, an entropic-regularized optimal transport is used (Cuturi, 2013).

$$\min_{\mathbf{\Pi} \in \boldsymbol{U}(\mu,\nu)} \quad \sum_{i=1}^{n} \sum_{j=1}^{m} \pi_{ij} c(\boldsymbol{x}_i, \boldsymbol{y}_j) + \epsilon H(\mathbf{\Pi}),$$

where $\epsilon \geq 0$ is the regularization parameter, and $H(\mathbf{\Pi}) = \sum_{i=1}^{n} \sum_{j=1}^{m} \pi_{ij}(\log(\pi_{ij}) - 1)$ is the entropic regularization. If $\epsilon = 0$, then the regularized OT problem reduces to the EMD problem. Owing to entropic regularization, the entropic regularized OT problem can be accurately solved using Sinkhorn iteration (Cuturi, 2013) with a $O(nm)$ computational cost (See Algorithm 1).

**Wasserstein distance:** If the cost function is defined as $c(\boldsymbol{x}, \boldsymbol{y}) = d(\boldsymbol{x}, \boldsymbol{y})$ with $d(\boldsymbol{x}, \boldsymbol{y})$ as a distance function and $p \geq 1$, then we define the $p$-Wasserstein distance of two discrete measures $\mu = \sum_{i=1}^{n} a_i \delta_{\boldsymbol{x}_i}$ and $\nu = \sum_{j=1}^{m} b_j \delta_{\boldsymbol{y}_j}$ as

$$W_p(\mu, \nu) = \left( \min_{\mathbf{\Pi} \in \boldsymbol{U}(\mu,\nu)} \sum_{i=1}^{n} \sum_{j=1}^{m} \pi_{ij} d(\boldsymbol{x}_i, \boldsymbol{y}_j)^p \right)^{1/p}.$$

Recently, a robust variant of the Wasserstein distance, called the subspace robust Wasserstein distance (SRW), was proposed (Paty & Cuturi, 2019). The SRW computes the OT problem in the discriminative subspace. This can be determined by solving dimensionality-reduction problems. Owing to the robustness, it can compute the Wasserstein from noisy data. The SRW is given as

$$\text{SRW}(\mu, \nu) = \left( \min_{\mathbf{\Pi} \in \boldsymbol{U}(\mu,\nu)} \max_{\boldsymbol{U} \in \mathbb{R}^{d \times k}, \boldsymbol{U}^\top \boldsymbol{U} = \boldsymbol{I}_k} \sum_{i=1}^{n} \sum_{j=1}^{m} \pi_{ij} \|\boldsymbol{U}^\top \boldsymbol{x}_i - \boldsymbol{U}^\top \boldsymbol{y}_j\|_2^2 \right)^{\frac{1}{2}}, \tag{2}$$

where $\boldsymbol{U}$ is the projection matrix with $k \leq d$, and $\boldsymbol{I}_k \in \mathbb{R}^{k \times k}$ is the identity matrix. The SRW or its relaxed problem can be efficiently estimated using either eigenvalue decomposition or the Frank–Wolfe algorithm.

## 3 PROPOSED METHOD

This paper proposes FROT. We assume that the vectors are grouped as $\boldsymbol{x} = (\boldsymbol{x}^{(1)\top}, \ldots, \boldsymbol{x}^{(L)\top})^\top$ and $\boldsymbol{y} = (\boldsymbol{y}^{(1)\top}, \ldots, \boldsymbol{y}^{(L)\top})^\top$. Here, $\boldsymbol{x}^{(\ell)} \in \mathbb{R}^{d_\ell}$ and $\boldsymbol{y}^{(\ell)} \in \mathbb{R}^{d_\ell}$ are the $d_\ell$ dimensional vectors, where $\sum_{\ell=1}^{L} d_\ell = d$. This setting is useful if we know the explicit group structure for the feature vectors a priori. In an application in $L$-layer neural networks, we consider $\boldsymbol{x}^{(\ell)}$ and $\boldsymbol{y}^{(\ell)}$ as outputs of the $\ell$th layer of the network. If we do not have a priori information, we can consider each feature independently (i.e., $d_1 = d_2 = \ldots = d_L = 1$ and $L = d$). All proofs in this section are provided in the Appendix.

### 3.1 FEATURE-ROBUST OPTIMAL TRANSPORT (FROT)

The FROT formulation is given by

$$\min_{\mathbf{\Pi} \in \boldsymbol{U}(\mu,\nu)} \max_{\boldsymbol{\alpha} \in \boldsymbol{\Sigma}^L} \sum_{i=1}^{n} \sum_{j=1}^{m} \pi_{ij} \sum_{\ell=1}^{L} \alpha_\ell c(\boldsymbol{x}_i^{(\ell)}, \boldsymbol{y}_j^{(\ell)}), \tag{3}$$

where $\mathbf{\Sigma}^L = \{\boldsymbol{\alpha} \in \mathbb{R}_+^L : \boldsymbol{\alpha}^\top \mathbf{1}_L = 1\}$ is the probability simplex. The underlying concept of FROT is to estimate the transport plan $\mathbf{\Pi}$ using distinct groups with large distances between $\{\boldsymbol{x}_i^{(\ell)}\}_{i=1}^n$ and $\{\boldsymbol{y}_j^{(\ell)}\}_{j=1}^m$. We note that determining the transport plan in nondistinct groups is difficult because the data samples in $\{\boldsymbol{x}_i^{(\ell)}\}_{i=1}^n$ and $\{\boldsymbol{y}_j^{(\ell)}\}_{j=1}^m$ overlap. By contrast, in distinct groups, $\{\boldsymbol{x}_i^{(\ell)}\}_{i=1}^n$ and $\{\boldsymbol{y}_j^{(\ell)}\}_{j=1}^m$ are different, and this aids in determining an optimal transport plan. This is an intrinsically similar idea to the subspace robust Wasserstein distance (Paty & Cuturi, 2019), which estimates the transport plan in the discriminative subspace, while our approach selects important groups. Therefore, FROT can be regarded as a feature selection variant of the vanilla OT problem in Eq. (1), whereas the subspace robust version uses dimensionality-reduction counterparts.

---

**Algorithm 1** Sinkhorn algorithm.

1: **Input:** $\boldsymbol{a}, \boldsymbol{b}, \boldsymbol{C}, \epsilon, t_{max}$
2: Initialize $\boldsymbol{K} = e^{-\boldsymbol{C}/\epsilon}, \boldsymbol{u} = \mathbf{1}_n, \boldsymbol{v} = \mathbf{1}_m, t = 0$
3: **while** $t \leq t_{max}$ and not converge **do**
4:     $\boldsymbol{u} = \boldsymbol{a}/(\boldsymbol{K}\boldsymbol{v})$
5:     $\boldsymbol{v} = \boldsymbol{b}/(\boldsymbol{K}^\top \boldsymbol{u})$
6:     $t = t + 1$
7: **end while**
8: **return** $\mathbf{\Pi} = \text{diag}(\boldsymbol{u})\boldsymbol{K}\text{diag}(\boldsymbol{v})$

---

**Algorithm 2** FROT with the Frank–Wolfe.

1: **Input:** $\{\boldsymbol{x}_i\}_{i=1}^n, \{\boldsymbol{y}_j\}_{j=1}^m, \eta$, and $\epsilon$.
2: Initialize $\mathbf{\Pi}$, compute $\{\boldsymbol{C}_\ell\}_{\ell=1}^L$.
3: **for** $t = 0 \ldots T$ **do**
4:     $\widehat{\mathbf{\Pi}} = \text{argmin}_{\mathbf{\Pi}\in\boldsymbol{U}(\mu,\nu)}\langle\mathbf{\Pi}, \boldsymbol{M}_{\mathbf{\Pi}^{(t)}}\rangle + \epsilon H(\mathbf{\Pi})$
5:     $\mathbf{\Pi}^{(t+1)} = (1-\gamma)\mathbf{\Pi}^{(t)} + \gamma\widehat{\mathbf{\Pi}}$
6:     with $\gamma = \frac{2}{2+t}$.
7: **end for**
8: **return** $\mathbf{\Pi}^{(T)}$

---

Using FROT, we can define a $p$-feature robust Wasserstein distance ($p$-FRWD).

**Proposition 1** *For the distance function $d(\boldsymbol{x}, \boldsymbol{y})$,*

$$\text{FRWD}_p(\mu, \nu) = \left( \min_{\mathbf{\Pi}\in\boldsymbol{U}(\mu,\nu)} \max_{\boldsymbol{\alpha}\in\mathbf{\Sigma}^L} \quad \sum_{i=1}^n \sum_{j=1}^m \pi_{ij} \sum_{\ell=1}^L \alpha_\ell d(\boldsymbol{x}_i^{(\ell)}, \boldsymbol{y}_j^{(\ell)})^p \right)^{1/p}, \quad (4)$$

*is a distance for $p \geq 1$.*

Note that we can show that 2-FRWD is a special case of SRW with $d(\boldsymbol{x}, \boldsymbol{y}) = \|\boldsymbol{x} - \boldsymbol{y}\|_2$ (See Appendix). The key difference between SRW and FRWD is that FRWD can use any distance, while SRW can only use $d(\boldsymbol{x}, \boldsymbol{y}) = \|\boldsymbol{x} - \boldsymbol{y}\|_2$.

## 3.2 FROT OPTIMIZATION

Here, we propose two FROT algorithms based on the Frank–Wolfe algorithm and linear programming.

**Frank–Wolfe:** We propose a continuous variant of the FROT algorithm using the Frank–Wolfe algorithm, which can be fully differentiable. To this end, we introduce entropic regularization for $\boldsymbol{\alpha}$ and rewrite the FROT as a function of $\mathbf{\Pi}$. Therefore, we solve the following problem for $\boldsymbol{\alpha}$:

$$\min_{\mathbf{\Pi}\in\boldsymbol{U}(\mu,\nu)} \max_{\boldsymbol{\alpha}\in\mathbf{\Sigma}^L} \quad J_\eta(\mathbf{\Pi}, \boldsymbol{\alpha}), \text{with } J_\eta(\mathbf{\Pi}, \boldsymbol{\alpha}) = \sum_{i=1}^n \sum_{j=1}^m \pi_{ij} \sum_{\ell=1}^L \alpha_\ell c(\boldsymbol{x}_i^{(\ell)}, \boldsymbol{y}_j^{(\ell)}) - \eta H(\boldsymbol{\alpha}),$$

where $\eta \geq 0$ is the regularization parameter, and $H(\boldsymbol{\alpha}) = \sum_{\ell=1}^L \alpha_\ell(\log(\alpha_\ell) - 1)$ is the entropic regularization for $\boldsymbol{\alpha}$. An advantage of entropic regularization is that the nonnegative constraint is naturally satisfied, and the entropic regularizer is a strong convex function.

**Lemma 2** *The optimal solution of the optimization problem*

$$\boldsymbol{\alpha}^* = \underset{\boldsymbol{\alpha}\in\mathbf{\Sigma}^L}{\text{argmax}} \quad J_\eta(\mathbf{\Pi}, \boldsymbol{\alpha}), \text{ with } J_\eta(\mathbf{\Pi}, \boldsymbol{\alpha}) = \sum_{\ell=1}^L \alpha_\ell \phi_\ell - \eta H(\boldsymbol{\alpha})$$

*with a fixed admissible transport plan* $\mathbf{\Pi} \in \mathbf{U}(\mu, \nu)$, *is given by*

$$\alpha_\ell^* = \frac{\exp\left(\frac{1}{\eta}\phi_\ell\right)}{\sum_{\ell'=1}^{L} \exp\left(\frac{1}{\eta}\phi_{\ell'}\right)} \text{ with } J_\eta(\mathbf{\Pi}, \boldsymbol{\alpha}^*) = \eta \log\left(\sum_{\ell=1}^{L} \exp\left(\frac{1}{\eta}\phi_\ell\right)\right) + \eta.$$

Using Lemma 2 (or Lemma 4 in Nesterov (2005)) together with the setting $\phi_\ell = \sum_{i=1}^{n}\sum_{j=1}^{m} \pi_{ij} c(\boldsymbol{x}_i^{(\ell)}, \boldsymbol{y}_i^{(\ell)}) = \langle \mathbf{\Pi}, \boldsymbol{C}_\ell \rangle$, $[\boldsymbol{C}_\ell]_{ij} = c(\boldsymbol{x}_i^{(\ell)}, \boldsymbol{y}_i^{(\ell)})$, the global problem is equivalent to

$$\min_{\mathbf{\Pi} \in \boldsymbol{U}(\mu, \nu)} G_\eta(\mathbf{\Pi}), \text{ with } G_\eta(\mathbf{\Pi}) = \eta \log\left(\sum_{\ell=1}^{L} \exp\left(\frac{1}{\eta}\langle\mathbf{\Pi}, \boldsymbol{C}_\ell\rangle\right)\right). \tag{5}$$

Note that this is known as a smoothed max-operator (Nesterov, 2005; Blondel et al., 2018). Specifically, regularization parameter $\eta$ controls the "smoothness" of the maximum.

**Proposition 3** $G_\eta(\mathbf{\Pi})$ *is a convex function relative to* $\mathbf{\Pi}$.

The derived optimization problem of FROT is convex. Therefore, we can determine globally optimal solutions. Note that the SRW optimization problem is not jointly convex (Paty & Cuturi, 2019) for the projection matrix and the transport plan. In this study, we employ the Frank–Wolfe algorithm (Frank & Wolfe, 1956; Jaggi, 2013), using which we approximate $G_\eta(\mathbf{\Pi})$ with linear functions at $\mathbf{\Pi}^{(t)}$ and move $\mathbf{\Pi}$ toward the optimal solution in the convex set (See Algorithm 2).

The derivative of the loss function $G_\eta(\mathbf{\Pi})$ at $\mathbf{\Pi}^{(t)}$ is given by

$$\left.\frac{\partial G_\eta(\mathbf{\Pi})}{\partial \mathbf{\Pi}}\right|_{\mathbf{\Pi}=\mathbf{\Pi}^{(t)}} = \sum_{\ell=1}^{L} \alpha_\ell^{(t)} \boldsymbol{C}_\ell = \boldsymbol{M}_{\mathbf{\Pi}^{(t)}} \text{ with } \alpha_\ell^{(t)} = \frac{\exp\left(\frac{1}{\eta}\langle\mathbf{\Pi}^{(t)}, \boldsymbol{C}_\ell\rangle\right)}{\sum_{\ell'=1}^{L} \exp\left(\frac{1}{\eta}\langle\mathbf{\Pi}^{(t)}, \boldsymbol{C}_{\ell'}\rangle\right)}.$$

Then, we update the transport plan by solving the EMD problem:

$$\mathbf{\Pi}^{(t+1)} = (1-\gamma)\mathbf{\Pi}^{(t)} + \gamma\widehat{\mathbf{\Pi}} \text{ with } \widehat{\mathbf{\Pi}} = \operatorname*{argmin}_{\mathbf{\Pi} \in \boldsymbol{U}(\mu, \nu)} \langle \mathbf{\Pi}, \boldsymbol{M}_{\mathbf{\Pi}^{(t)}} \rangle,$$

where $\gamma = 2/(2+k)$. Note that $\boldsymbol{M}_{\mathbf{\Pi}^{(t)}}$ is given by the weighted sum of the cost matrices. Thus, we can utilize multiple features to estimate the transport plan $\mathbf{\Pi}$ for the relaxed problem in Eq. (5).

Using the Frank–Wolfe algorithm, we can obtain the optimal solution. However, solving the EMD problem requires a cubic computational cost that can be expensive if $n$ and $m$ are large. To address this, we can solve the regularized OT problem, which requires $O(nm)$. We denote the Frank–Wolfe algorithm with EMD as FW-EMD and the Frank–Wolfe algorithm with Sinkhorn as FW-Sinkhorn.

**Computational complexity:** The proposed method depends on the Sinkhorn algorithm, which requires an $O(nm)$ operation. The computation of the cost matrix in each subproblem needs an $O(Lnm)$ operation, where $L$ is the number of groups. Therefore, the entire complexity is $O(TLnm)$, where $T$ is the number of Frank–Wolfe iterations (in general, $T = 10$ is sufficient).

**Proposition 4** *For each* $t \geq 1$, *the iteration* $\mathbf{\Pi}^{(t)}$ *of Algorithm 2 satisfies*

$$G_\eta(\mathbf{\Pi}^{(t)}) - G_\eta(\mathbf{\Pi}^*) \leq \frac{4\sigma_{max}(\boldsymbol{\Phi}^\top\boldsymbol{\Phi})}{\eta(t+2)}(1+\delta),$$

*where* $\sigma_{max}(\boldsymbol{\Phi}^\top\boldsymbol{\Phi})$ *is the largest eigenvalue of the matrix* $\boldsymbol{\Phi}^\top\boldsymbol{\Phi}$ *and* $\boldsymbol{\Phi} = (\text{vec}(\boldsymbol{C}_1), \text{vec}(\boldsymbol{C}_2), \ldots, \text{vec}(\boldsymbol{C}_L))^\top$; *and* $\delta \geq 0$ *is the accuracy to which internal linear subproblems are solved.*

Based on Proposition 4, the number of iterations depends on $\eta$, $\epsilon$, and the number of groups. If we set a small $\eta$, convergence requires more time. In addition, if we use entropic regularization with a large $\epsilon$, the $\delta$ in Proposition 4 can be large. Finally, if we use more groups, the largest eigenvalue of the matrix $\boldsymbol{\Phi}^\top\boldsymbol{\Phi}$ can be larger. Note that the constant term of the upper bound is large; however, the Frank–Wolfe algorithm converges quickly in practice.

**Linear Programming:** Because $\lim_{\eta \to 0^+} G_\eta(\mathbf{\Pi}) = \max_{\ell \in \{1,2,\ldots,L\}} \sum_{i=1}^n \sum_{j=1}^m \pi_{ij} c(\boldsymbol{x}_i^{(\ell)}, \boldsymbol{y}_j^{(\ell)})$, the FROT problem can also be written as

$$\min_{\mathbf{\Pi} \in \boldsymbol{U}(\mu,\nu)} \max_{\ell \in \{1,2,\ldots,L\}} \sum_{i=1}^n \sum_{j=1}^m \pi_{ij} c(\boldsymbol{x}_i^{(\ell)}, \boldsymbol{y}_j^{(\ell)}). \tag{6}$$

Because the objective is the max of linear functions, it is convex with respect to $\mathbf{\Pi}$. We can solve the problem via linear programming:

$$\min_{\mathbf{\Pi} \in \boldsymbol{U}(\mu,\nu),t} t, \quad \text{s.t.} \quad \langle \mathbf{\Pi}, \boldsymbol{C}_\ell \rangle \leq t, \ell = 1,2,\ldots,L. \tag{7}$$

This optimization can be easily solved using an off-the-shelf LP package. However, the computational cost of this LP problem is high in general (i.e., $O(n^3), n = m$).

### 3.3 Application: Semantic Correspondence

We applied our proposed FROT algorithm to semantic correspondence. The semantic correspondence is a problem that determines the matching of objects in two images. That is, given input image pairs $(A, B)$, with common objects, we formulated the semantic correspondence problem to estimate the transport plan from the key points in $A$ to those in $B$; this framework was proposed in (Liu et al., 2020). In Figure 2, we show an overview of our proposed framework.

**Cost matrix computation $\boldsymbol{C}_\ell$:** In our framework, we employed a pretrained convolutional neural network to extract dense feature maps for each convolutional layer. The dense feature map of the $\ell$th layer output of the $s$th image is given by

$$\boldsymbol{f}_{s,q+(r-1)h_s}^{(\ell,s)} \in \mathbb{R}^{d_\ell}, \ q = 1,2,\ldots,h_s, r = 1,2,\ldots,w_s, \ell = 1,2,\ldots,L,$$

where $w_s$ and $h_s$ are the width and height of the $s$th image, respectively, and $d_\ell$ is the dimension of the $\ell$th layer's feature map. Note that because the dimension of the dense feature map is different for each layer, we sample feature maps to the size of the 1st layer's feature map size (i.e., $h_s \times w_s$).

The $\ell$th layer's cost matrix for images $s$ and $s'$ is given by

$$[\boldsymbol{C}_\ell]_{ij} = \|\boldsymbol{f}_i^{(\ell,s)} - \boldsymbol{f}_j^{(\ell,s')}\|_2^2, \ i = 1,2,\ldots,w_s h_s, \ j = 1,2,\ldots,w_{s'} h_{s'}.$$

A potential problem with FROT is that the estimation depends significantly on the magnitude of the cost of each layer (also known as a group). Hence, normalizing each cost matrix is important. Therefore, we normalized each feature vector by $\boldsymbol{f}_i^{(\ell,s)} \leftarrow \boldsymbol{f}_i^{(\ell,s)}/\|\boldsymbol{f}_i^{(\ell,s)}\|_2$. Consequently, the cost matrix is given by $[\boldsymbol{C}_\ell]_{ij} = 2 - 2\boldsymbol{f}_i^{(\ell,s)^\top} \boldsymbol{f}_j^{(\ell,s')}$. We can use distances such as the $L1$ distance.

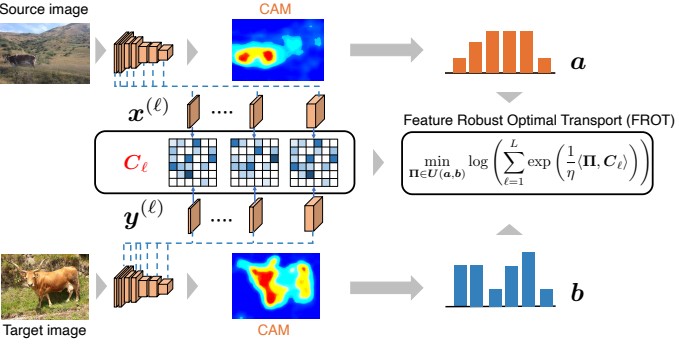

**Computation of $\boldsymbol{a}$ and $\boldsymbol{b}$ with staircase re-weighting:** For semantic correspondence, setting $\boldsymbol{a} \in \mathbb{R}^{h_s w_s}$ and $\boldsymbol{b} \in \mathbb{R}^{h_{s'} w_{s'}}$ is important because semantic correspondence can be affected by background clutter. Therefore, we generated the class activation maps (Zhou et al., 2016) for the source and target images and used them as $\boldsymbol{a}$ and $\boldsymbol{b}$, respectively. For CAM, we chose the class with the highest classification probability and normalized it to the range $[0, 1]$.

Figure 2: Semantic correspondence framework based on FROT.

## 4 Related Work

**OT algorithms:** The Wasserstein distance can be determined by solving the OT problem. An advantage of the Wasserstein distance is its robustness to noise; moreover, we can obtain the transport

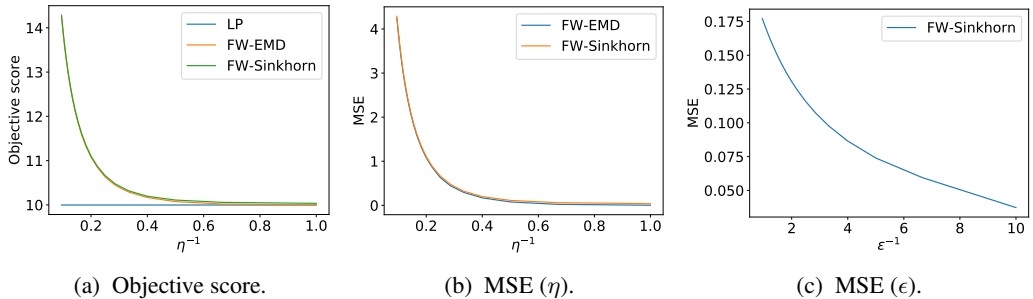

(a) Objective score.    (b) MSE ($\eta$).    (c) MSE ($\epsilon$).

Figure 3: (a) Objective scores for LP, FW-EMD, and FW-Sinkhorn. (b) MSE between transport plan of LP and FW-EMD and that with LP and FW-Sinkhorn with different $\eta$. (c) MSE between transport plan of LP and FW-Sinkhorn with different $\epsilon$.

plan, which is useful for many machine learning applications. To reduce the computation cost for the Wasserstein distance, the sliced Wasserstein distance is useful (Kolouri et al., 2016). Recently, a tree variant of the Wasserstein distance was proposed (Evans & Matsen, 2012; Le et al., 2019; Sato et al., 2020); the sliced Wasserstein distance is a special case of this algorithm.

The approach most closely related to FROT is a robust variant of the Wasserstein distance, called the subspace robust Wasserstein distance (SRW) (Paty & Cuturi, 2019). SRW computes the OT problem in a discriminative subspace; this is possible by solving dimensionality-reduction problems. Owing to the robustness, SRW can successfully compute the Wasserstein distance from noisy data. The max–sliced Wasserstein distance (Deshpande et al., 2019) and its generalized counterpart (Kolouri et al., 2019) can also be regarded as subspace-robust Wasserstein methods. Note that SRW (Paty & Cuturi, 2019) is a *min–max* based approach, while the max–sliced Wasserstein distances (Deshpande et al., 2019; Kolouri et al., 2019) are *max–min* approaches. The FROT is a feature selection variant of the Wasserstein distance, whereas the subspace approaches are used for dimensionality reduction.

As a parallel work, a general minimax optimal transport problem called the robust Kantorovich problem (RKP) was recently proposed (Dhouib et al., 2020). RKP involves using a cutting-set method for a general minmax optimal transport problem that includes the FROT problem as a special case. The approaches are technically similar; however, our problem and that of Dhouib et al. (2020) are intrinsically different. Specifically, we aim to solve a high-dimensional OT problem using feature selection and apply it to semantic correspondence problems, while the RKP approach focuses on providing a general framework and uses it for color transformation problems. As a technical difference, the cutting-set method may not converge to an optimal solution if we use the regularized OT (Dhouib et al., 2020). By contrast, because we use a Frank–Wolfe algorithm, our algorithm converges to a true objective function with regularized OT solvers. The multiobjective optimal transport (MOT) is an approach (Scetbon et al., 2020) parallel to ours. The key difference between FROT and MOT is that MOT tries to use the weighted sum of cost functions, while FROT considers the worst case. Moreover, as applications, we focus on the cost matrices computed from subsets of features, while MOT considers cost matrices with different distance functions.

## 5 EXPERIMENTS

### 5.1 SYNTHETIC DATA

We compare FROT with a standard OT using synthetic datasets. In these experiments, we initially generate two-dimensional vectors $\boldsymbol{x} \sim N(\boldsymbol{\mu}_x, \boldsymbol{\Sigma}_x)$ and $\boldsymbol{y} \sim N(\boldsymbol{\mu}_y, \boldsymbol{\Sigma}_y)$. Here, we set $\boldsymbol{\mu}_x = (-5, 0)^\top$, $\boldsymbol{\mu}_y = (5, 0)^\top$, $\boldsymbol{\Sigma}_x = \boldsymbol{\Sigma}_y = ((5, 1)^\top, (4, 1)^\top)$. Then, we concatenate $\boldsymbol{z}_x \sim N(\boldsymbol{0}_8, \boldsymbol{I}_8)$ and $\boldsymbol{z}_y \sim N(\boldsymbol{0}_8, \boldsymbol{I}_8)$ to $\boldsymbol{x}$ and $\boldsymbol{y}$, respectively, to give $\widetilde{\boldsymbol{x}} = (\boldsymbol{x}^\top, \boldsymbol{z}_x^\top)$, $\widetilde{\boldsymbol{y}} = (\boldsymbol{y}^\top, \boldsymbol{z}_y^\top)$.

For FROT, we set $\eta = 1.0$ and the number of iterations of the Frank–Wolfe algorithm as $T = 10$. The regularization parameter is set to $\epsilon = 0.02$ for all methods. To show the proof-of-concept, we set the true features as a group and the remaining noise features as another group.

Fig. 1a shows the correspondence from $\boldsymbol{x}$ and $\boldsymbol{y}$ with the vanilla OT algorithm. Figs. 1b and 1c show the correspondence of FROT and OT with $\widetilde{\boldsymbol{x}}$ and $\widetilde{\boldsymbol{y}}$, respectively. Although FROT can identify

Table 1: Per-class PCK ($\alpha_{bbox} = 0.1$) results using SPair-71k. All models use ResNet101. The numbers in the bracket of SRW are the input layer indicies.

| | Methods | aero | bike | bird | boat | bottle | bus | car | cat | chair | cow | dog | horse | moto | person | plant | sheep | train | tv | all |
|---|---|---|---|---|---|---|---|---|---|---|---|---|---|---|---|---|---|---|---|---|
| SPair-71k finetuned models | CNNGeo (Rocco et al., 2017) | 23.4 | 16.7 | 40.2 | 14.3 | 36.4 | 27.7 | 26.0 | 32.7 | 12.7 | 27.4 | 22.8 | 13.7 | 20.9 | 21.0 | 17.5 | 10.2 | 30.8 | 34.1 | 20.6 |
| | A2Net (Hongsuck Seo et al., 2018) | 22.6 | 18.5 | 42.0 | 16.4 | 37.9 | **30.8** | 26.5 | 35.6 | 13.3 | 29.6 | 24.3 | 16.0 | 21.6 | 22.8 | 20.5 | 13.5 | 31.4 | 36.5 | 22.3 |
| | WeakAlign (Rocco et al., 2018a) | 22.2 | 17.6 | 41.9 | 15.1 | 38.1 | 27.4 | **27.2** | 31.8 | 12.8 | 26.8 | 22.6 | 14.2 | 20.0 | 22.2 | 17.9 | 10.4 | 32.2 | 35.1 | 20.9 |
| | NC-Net (Rocco et al., 2018b) | 17.9 | 12.2 | 32.1 | 11.7 | 29.0 | 19.9 | 16.1 | 39.2 | 9.9 | 23.9 | 18.8 | 15.7 | 17.4 | 15.9 | 14.8 | 9.6 | 24.2 | 31.1 | 20.1 |
| SPair-71k validation | HPF (Min et al., 2019a) | 25.2 | 18.9 | 52.1 | 15.7 | 38.0 | 22.8 | 19.1 | 52.9 | 17.9 | 33.0 | 32.8 | 20.6 | 24.4 | 27.9 | 21.1 | 15.9 | 31.5 | 35.6 | 28.2 |
| | OT-HPF (Liu et al., 2020) | 32.6 | 18.9 | **62.5** | 20.7 | 42.0 | 26.1 | 20.4 | 61.4 | **19.7** | **41.3** | **41.7** | 29.8 | **29.6** | **31.8** | 25.0 | 23.5 | 44.7 | 37.0 | 33.9 |
| | FROT($\eta = 0.2, \epsilon = 0.4$) | 35.1 | 20.3 | 59.8 | 21.1 | **42.9** | 27.7 | 21.2 | **63.5** | 18.8 | 39.7 | 37.9 | 29.2 | 28.8 | 29.9 | **28.2** | 24.3 | 52.1 | 39.5 | **34.7** |
| Without SPair-71k validation | OT | 30.1 | 16.5 | 50.4 | 17.3 | 38.0 | 22.9 | 19.7 | 54.3 | 17.0 | 28.4 | 31.3 | 22.1 | 28.0 | 19.5 | 21.0 | 17.8 | 42.6 | 28.8 | 28.3 |
| | FROT ($\eta = 0.3, T = 3$) | 35.0 | **20.9** | 56.3 | **23.4** | 40.7 | 27.2 | 21.9 | 62.0 | 17.5 | 38.8 | 36.2 | 27.9 | 28.0 | 30.4 | 26.9 | 23.1 | 49.7 | 38.4 | 33.7 |
| | FROT ($\eta = 0.3, T = 10$) | 34.9 | **20.9** | 56.4 | **23.4** | 40.7 | 27.2 | 22.0 | 62.0 | 17.5 | 38.8 | 36.2 | 27.8 | 28.2 | 30.2 | 26.9 | 22.9 | 49.7 | 38.5 | 33.7 |
| | FROT ($\eta = 0.5, T = 3$) | 34.1 | 18.8 | 56.9 | 19.9 | 40.0 | 25.6 | 19.2 | 61.9 | 17.4 | 38.7 | 36.5 | 25.6 | 26.9 | 27.2 | 26.3 | 22.1 | 50.3 | 38.6 | 32.8 |
| | FROT ($\eta = 0.5, T = 10$) | 34.0 | 18.9 | 57.0 | 19.9 | 40.0 | 25.6 | 19.2 | 61.9 | 17.3 | 38.8 | 36.5 | 25.6 | 26.8 | 27.4 | 26.4 | 22.1 | 50.3 | 38.8 | 32.8 |
| | FROT ($\eta = 0.7, T = 3$) | 33.4 | 19.4 | 56.6 | 20.0 | 39.6 | 26.1 | 19.1 | 62.4 | 17.9 | 38.0 | 36.5 | 26.0 | 27.5 | 26.5 | 25.5 | 21.6 | 49.7 | 38.9 | 32.7 |
| | FROT ($\eta = 0.7, T = 10$) | 33.3 | 19.5 | 56.6 | 19.9 | 39.5 | 26.0 | 19.1 | 62.4 | 17.9 | 38.0 | 36.5 | 26.0 | 27.4 | 26.5 | 25.6 | 21.6 | 49.6 | 38.9 | 32.7 |
| | SRW (layers = {1, 32–34}) | 29.4 | 14.0 | 43.7 | 15.6 | 33.8 | 21.0 | 17.6 | 48.0 | 12.9 | 23.3 | 26.5 | 19.8 | 25.5 | 17.6 | 16.7 | 15.2 | 37.1 | 20.5 | 24.5 |
| | SRW (layers = {1, 31–34}) | 29.7 | 14.3 | 44.3 | 15.7 | 34.2 | 21.3 | 17.8 | 48.5 | 13.1 | 23.6 | 27.1 | 20.0 | 25.8 | 18.1 | 16.9 | 15.2 | 37.3 | 21.0 | 24.8 |
| | SRW (layers = {1, 30–34}) | 29.8 | 14.7 | 45.6 | 15.9 | 34.8 | 21.5 | 18.0 | 49.3 | 13.3 | 24.0 | 27.7 | 20.6 | 25.7 | 18.7 | 17.2 | 15.3 | 37.7 | 21.5 | 25.2 |
| | FROT (layers = {1, 32–34}) | 32.3 | 15.7 | 43.1 | 18.4 | 30.7 | 22.5 | 20.6 | 44.5 | 10.3 | 23.1 | 23.9 | 19.5 | 23.6 | 22.0 | 14.7 | 15.3 | 37.4 | 18.0 | 24.3 |
| | FROT (layers = {1, 31–34}) | 35.3 | 16.5 | 45.3 | 20.5 | 33.0 | 25.0 | 21.6 | 48.1 | 11.4 | 25.9 | 26.9 | 22.4 | 25.2 | 25.0 | 16.5 | 17.1 | 40.5 | 21.2 | 26.6 |
| | FROT (layers = {1, 30–34}) | **36.7** | 18.1 | 48.8 | 22.3 | 34.5 | 27.5 | 23.0 | 51.3 | 12.9 | 28.4 | 30.3 | 24.2 | 26.4 | 27.3 | 19.5 | 18.1 | 43.4 | 24.9 | 28.8 |

a suitable matching, the OT fails to obtain a significant correspondence. We observed that the $\alpha$ parameter corresponding to a true group is $\alpha_1 = 0.9999$. Moreover, we compared the objective scores of the FROT with LP, FW-EMD, and FW-Sinkhorn ($\epsilon = 0.1$). Figure 3a shows the objective scores of FROTs with the different solvers, and both FW-EMD and FW-Sinkhorn can achieve almost the same objective score with a relatively small $\eta$. Moreover, Figure 3b shows the mean squared error between the LP method and the FW counterparts. Similar to the objective score cases, it can yield a similar transport plan with a relatively small $\eta$. Finally, we evaluated the FW-Sinkhorn by changing the regularization parameter $\eta$. In this experiment, we set $\eta = 1$ and varied the $\epsilon$ values. The result shows that we can obtain an accurate transport plan with a relatively small $\epsilon$.

## 5.2 SEMANTIC CORRESPONDENCE

We evaluated our FROT algorithm for semantic correspondence. In this study, we used the SPair-71k (Min et al., 2019b). The SPair-71k dataset consists of $70,958$ image pairs with variations in viewpoint and scale. For evaluation, we employed a percentage of accurate key points (PCK), which counts the number of accurately predicted key points given a fixed threshold (Min et al., 2019b). All semantic correspondence experiments were run on a Linux server with NVIDIA P100.

For the optimal transport based frameworks, we employed ResNet101 (He et al., 2016) pretrained on ImageNet (Deng et al., 2009) for feature and activation map extraction. The ResNet101 consists of 34 convolutional layers and the entire number of features is $d = 32,576$. Note that we did not fine-tune the network. We compared the proposed method with several baselines (Min et al., 2019b) and the SRW[1]. Owing to the computational cost and the required memory size for SRW, we used the first and the last few convolutional layers of ResNet101 as the input of SRW. In our experiments, we empirically set $T = 3$ and $\epsilon = 0.1$ for FROT and SRW, respectively. For SRW, we set the number of latent dimension as $k = 50$ for all experiments. HPF (Min et al., 2019a) and OT-HPF (Liu et al., 2020) are state-of-the-art methods for semantic correspondence. HPF and OT-HPF required the validation dataset to select important layers, whereas SRW and FROT did not require the validation dataset. OT is a simple optimal transport-based method that does not select layers.

Table 1 lists the per-class PCK results obtained using the SPair-71k dataset. FROT ($\eta = 0.3$) outperforms most existing baselines, including HPF and OT. Moreover, FROT ($\eta = 0.3$) is consistent with OT-HPF (Liu et al., 2020), which requires the validation dataset to select important layers. In this experiment, setting $\eta < 1$ results in favorable performance (See Table 3 in the Appendix). The computational costs of FROT is 0.29, while SRWs are 8.73, 11.73, 15.76, respectively. Surprisingly, FROT outperformed SRWs. However, this is mainly due to the used input layers. Therefore, scaling up SRW would be an interesting future work.

We further evaluated FROT by tuning hyperparameters $\eta$ and $\epsilon$ using validation sets, where the maximum search ranges for $\eta$ and $\epsilon$ are set to 0.2 to 2.0 and 0.1 to 0.6 with intervals of 0.1, respectively. Figure 6 in Appendix shows the average PCK scores for ($\eta, \epsilon$) pairs on the validation split of SPair-71k. By using hyperparameter search, we selected ($\eta = 0.2, \epsilon = 0.4$) as an optimal parameter. The FROT with optimal parameters outperforms the state-of-the-art method (Liu et al., 2020).

---

[1]https://github.com/francoispierrepaty/SubspaceRobustWasserstein

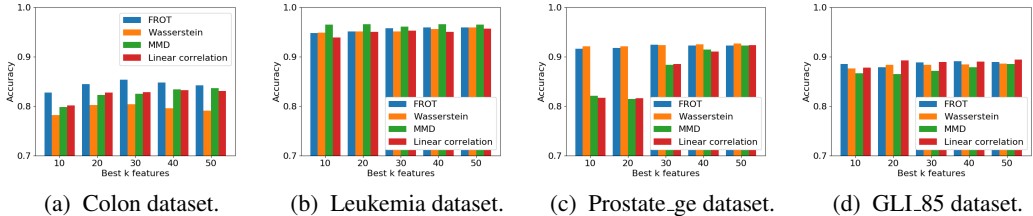

| (a) Colon dataset. | (b) Leukemia dataset. | (c) Prostate_ge dataset. | (d) GLI_85 dataset. |

Figure 4: Feature selection results. We average over 50 runs of accuracy (on test set) of SVM trained with top k features selected by several methods.

### 5.3 FEATURE SELECTION EXPERIMENTS

Since FROT finds the transport plan and discriminative features between $\boldsymbol{X}$ and $\boldsymbol{Y}$, we can use FROT as a feature-selection method. We considered $\boldsymbol{X} \in \mathbb{R}^{d \times n}$ and $\boldsymbol{Y} \in \mathbb{R}^{d \times m}$ as sets of samples from classes 1 and 2, respectively. The optimal important feature is given by

$$\widehat{\alpha}_\ell = \frac{\exp\left(\frac{1}{\eta}\langle\widehat{\boldsymbol{\Pi}}, \boldsymbol{C}_\ell\rangle\right)}{\sum_{\ell'=1}^d \exp\left(\frac{1}{\eta}\langle\widehat{\boldsymbol{\Pi}}, \boldsymbol{C}_{\ell'}\rangle\right)}, \text{ with } \widehat{\boldsymbol{\Pi}} = \operatorname*{argmin}_{\boldsymbol{\Pi} \in \boldsymbol{U}(\mu,\nu)} \quad \eta\log\left(\sum_{\ell=1}^d \exp\left(\frac{1}{\eta}\langle\boldsymbol{\Pi}, \boldsymbol{C}_\ell\rangle\right)\right),$$

where $[\boldsymbol{C}_\ell]_{ij} = (x_i^{(\ell)} - y_j^{(\ell)})^2$. Finally, we selected the top $K$ features by the ranking $\widehat{\boldsymbol{\alpha}}$. Hence, $\boldsymbol{\alpha}$ changes to a one-hot vector for a small $\eta$ and to $\alpha_k \approx \frac{1}{L}$ for a large $\eta$.

Here, we compared FROT with several baseline algorithms in terms of solving feature-selection problems. In this study, we employed a high-dimensional and a few sample datasets with two class classification tasks (see Table 2). All feature selection experiments were run on a Linux server with an Intel Xeon CPU E7-8890 v4 with 2.20 GHz and 2 TB RAM.

In our experiments, we initially randomly split the data into two sets (75% for training and 25% for testing) and used the training set for feature selection and building a classifier. Note that we standardized each feature using the training set. Then, we used the remaining set for the test. The trial was repeated 50 times, and we considered the averaged classification accuracy for all trials. Considered as baseline methods, we computed the Wasserstein distance, maximum mean discrepancy (MMD) (Gretton et al., 2007), and linear correlation[2] for each dimension and sorted them in descending order. Note that the Wasserstein distance is computed via sorting, which is computationally more efficient than the Sinkhorn algorithm when $d = 1$. Then, we selected the top $K$ features as important features. For FROT, we computed the feature importance and selected the features that had significant importance scores. In our experiments, we set $\eta = 1.0$ and $T = 10$. Then, we trained a two-class SVM[3] with the selected features.

Fig. 4 shows the average classification accuracy relative to the number of selected features. From Figure 4, FROT is consistent with the Wasserstein distance-based feature selection and outperforms the linear correlation method and the MMD for two datasets. Table 2 shows the computational time(s) of the methods. FROT is about two orders of magnitude faster than the Wasserstein distance and is also faster than MMD. Note that although MMD is as fast as the proposed method, it cannot determine the correspondence between samples.

## 6 CONCLUSION

In this paper, we proposed FROT for high-dimensional data. This approach jointly solves feature selection and OT problems. An advantage of FROT is that it is a convex optimization problem and can determine an accurate globally optimal solution using the Frank–Wolfe algorithm. We used FROT for high-dimensional feature selection and semantic correspondence problems. Through extensive experiments, we demonstrated that the proposed algorithm is consistent with state-of-the-art algorithms in both feature selection and semantic correspondence.

---

[2] https://scikit-learn.org/stable/modules/feature_selection.html
[3] https://scikit-learn.org/stable/modules/generated/sklearn.svm.SVC.html

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

APPENDIX

RELATED WORK

In addition to accelerating the computation, structured optimal transport incorporates structural information directly into OT problems (Alvarez-Melis et al., 2018). Specifically, they formulate the submodular optimal transport problem and solve the problem using a saddle-point mirror prox algorithm. Recently, more complex structured information was introduced in the OT problem, including the hierarchical structure (Alvarez-Melis et al., 2020; Yurochkin et al., 2019). These approaches successfully incorporate structured information into OT problems with respect to data samples. By contrast, FROT incorporates the structured information into features.

**OT applications:** OT has received significant attention for use in several computer vision tasks. Applications include Wasserstein distance estimation (Peyré et al., 2019), domain adaptation (Yan et al., 2018), multitask learning (Janati et al., 2019), barycenter estimation (Cuturi & Doucet, 2014), semantic correspondence (Liu et al., 2020), feature matching (Sarlin et al., 2019), photo album summarization (Liu et al., 2019), generative model (Arjovsky et al., 2017; Bunne et al., 2019), graph matching (Xu et al., 2019a;b), and the semantic correspondence (Liu et al., 2020).

PROOF OF PROPOSITION 1

For the distance function $d(\boldsymbol{x}, \boldsymbol{y})$, we prove that

$$\text{FRWD}_p(\mu, \nu) = \left( \min_{\boldsymbol{\Pi} \in \boldsymbol{U}(\mu,\nu)} \max_{\boldsymbol{\alpha} \in \boldsymbol{\Sigma}^L} \sum_{i=1}^n \sum_{j=1}^m \pi_{ij} \sum_{\ell=1}^L \alpha_\ell d(\boldsymbol{x}_i^{(\ell)}, \boldsymbol{y}_j^{(\ell)})^p \right)^{1/p}$$

is a distance for $p \geq 1$.

The symmetry can be read directly from the definition as we used distances that are symmetric. For the identity of indiscernibles, when $\text{FRWD}_p(\mu, \nu) = 0$ with the optimal $\boldsymbol{\alpha}$ and $\boldsymbol{\Pi}$, there exists $\ell$ such that $\alpha_\ell > 0$ (as $\boldsymbol{\alpha}$ is in the simplex set). As there is a max in the definition and $\sum_{ij} \pi_{ij}\alpha_\ell d(\boldsymbol{x}_i^{(\ell)}, \boldsymbol{y}_j^{(\ell)})^p = 0$, this means that $\forall \ell, \sum_{ij} \pi_{ij} d(\boldsymbol{x}_i^{(\ell)}, \boldsymbol{y}_j^{(\ell)})^p = 0$ and $\forall \ell, \mu^{(\ell)} = \nu^{(\ell)}$. Therefore, we have $\mu = \nu$ when $\text{FRWD}_p(\mu, \nu) = 0$.

When $\mu = \nu$, this means that $\boldsymbol{x}_i = \boldsymbol{y}_i$, $a_i = b_i, \forall\, i$, and $n = m$, and we have $d(\boldsymbol{x}_i, \boldsymbol{y}_j) = 0$ for $i = j$. Thus, for any $\alpha_\ell \geq 0$, the optimal transport plan is $\pi_{ii} > 0$ for $d(\boldsymbol{x}_i, \boldsymbol{y}_j) = 0$ and $\pi_{ij} = 0$ for $d(\boldsymbol{x}_i, \boldsymbol{y}_j) > 0$. Therefore, when $\mu = \nu$, we have $\text{FRWD}_p(\mu, \nu) = 0$.

TRIANGLE INEQUALITY

Let $\mu = \sum_{i=1}^n a_i \delta_{\boldsymbol{x}_i}$, $\nu = \sum_{j=1}^m b_j \delta_{\boldsymbol{y}_j}$, $\gamma = \sum_{k=1}^u c_k \delta_{\boldsymbol{z}_k}$ and $\boldsymbol{\alpha} \in \boldsymbol{\Sigma}^L$, we prove that

$$\text{FRWD}_p(\mu, \gamma) \leq \text{FRWD}_p(\mu, \nu) + \text{FRWD}_p(\nu, \gamma).$$

To simplify the notations in this proof, we define $\boldsymbol{D}_\ell$ as the distance "matrix" such that $[\boldsymbol{D}_\ell]_{ij} = d(\boldsymbol{x}_i^{(\ell)}, \boldsymbol{y}_j^{(\ell)})$ is the $i$th-row and $j$th-column element of the matrix $\boldsymbol{D}_\ell$, $[\boldsymbol{D}_\ell]_{jk} = d(\boldsymbol{y}_j^{(\ell)}, \boldsymbol{z}_k^{(\ell)})$, and $[\boldsymbol{D}_\ell]_{ik} = d(\boldsymbol{x}_i^{(\ell)}, \boldsymbol{z}_k^{(\ell)})$. Moreover, note that $\boldsymbol{D}_\ell^p$ is the "matrix," where each element is an element of $\boldsymbol{D}_\ell$ raised to the power $p$.

Consider that $\boldsymbol{P} \in \boldsymbol{U}(\mu, \nu)$ is the optimal transport plan of $\text{FRWD}_p(\mu, \nu)$, and $\boldsymbol{Q} \in \boldsymbol{U}(\nu, \gamma)$ is the optimal transport plan of $\text{FRWD}_p(\nu, \gamma)$, where $\gamma = \sum_{k=1}^r c_i \delta_{\boldsymbol{z}_i}$ is a discrete measure. Similar to the proof for the Wasserstein distance in (Peyré et al., 2019), let $\boldsymbol{S} = \boldsymbol{P}\text{diag}(1/\widetilde{\boldsymbol{b}})\boldsymbol{Q}$ with $\widetilde{\boldsymbol{b}}$ be a vector such that $\widetilde{b}_j = b_j$ if $b_j > 0$, and $b_j = 1$ otherwise. We can show that $\boldsymbol{S} \in \boldsymbol{U}(\mu, \gamma)$.

$$
\left(\min_{\boldsymbol{R}\in\boldsymbol{U}(\mu,\gamma)}\sum_{\ell=1}^{L}\alpha_\ell\langle\boldsymbol{R},\boldsymbol{D}_\ell^p\rangle\right)^{\frac{1}{p}} \leq \left(\sum_{\ell=1}^{L}\alpha_\ell\langle\boldsymbol{S},\boldsymbol{D}_\ell^p\rangle\right)^{\frac{1}{p}} = \left(\sum_{\ell=1}^{L}\alpha_\ell\sum_{ik}S_{ik}[\boldsymbol{D}_\ell]_{ik}^p\right)^{\frac{1}{p}}
$$

$$
\leq \left(\sum_{\ell=1}^{L}\alpha_\ell\sum_{ik}[\boldsymbol{D}_\ell]_{ik}^p\sum_{j}\frac{p_{ij}q_{jk}}{\widetilde{b}_j}\right)^{\frac{1}{p}} = \left(\sum_{\ell=1}^{L}\alpha_\ell\sum_{ijk}[\boldsymbol{D}_\ell]_{ik}^p\frac{p_{ij}q_{jk}}{\widetilde{b}_j}\right)^{\frac{1}{p}}
$$

$$
\leq \left(\sum_{\ell=1}^{L}\alpha_\ell\sum_{ijk}([\boldsymbol{D}_\ell]_{ij}+[\boldsymbol{D}_\ell]_{jk})^p\frac{p_{ij}q_{jk}}{\widetilde{b}_j}\right)^{\frac{1}{p}}
$$

By letting $g_{ijk\ell} = [\boldsymbol{D}_\ell]_{ij}(\alpha_\ell p_{ij}q_{jk}/\widetilde{b}_j)^{1/p}$ and $h_{ijk\ell} = [\boldsymbol{D}_\ell]_{ij}(\alpha_\ell p_{ij}q_{jk}/\widetilde{b}_j)^{1/p}$, the right-hand side of this inequality can be rewritten as

$$
\left(\sum_{\ell=1}^{L}\alpha_\ell\sum_{ijk}([\boldsymbol{D}_\ell]_{ij}+[\boldsymbol{D}_\ell]_{jk})^p\frac{p_{ij}q_{jk}}{\widetilde{b}_j}\right)^{\frac{1}{p}} = \left(\sum_{\ell=1}^{L}\sum_{ijk}(g_{ijk\ell}+h_{ijk\ell})^p\right)^{\frac{1}{p}}
$$

$$
\leq \left(\sum_{\ell=1}^{L}\sum_{ijk}g_{ijk\ell}^p\right)^{\frac{1}{p}} + \left(\sum_{\ell=1}^{L}\sum_{ijk}h_{ijk\ell}^p\right)^{\frac{1}{p}}
$$

$$
\leq \left(\sum_{\ell=1}^{L}\alpha_\ell\sum_{ijk}[\boldsymbol{D}_\ell]_{ij}^p\frac{p_{ij}q_{jk}}{\widetilde{b}_j}\right)^{\frac{1}{p}} + \left(\sum_{\ell=1}^{L}\alpha_\ell\sum_{ijk}[\boldsymbol{D}_\ell]_{jk}^p\frac{p_{ij}q_{jk}}{\widetilde{b}_j}\right)^{\frac{1}{p}}
$$

by the Minkovski inequality.

$$
\left(\min_{\boldsymbol{R}\in\boldsymbol{U}(\mu,\gamma)}\sum_{\ell=1}^{L}\alpha_\ell\langle\boldsymbol{R},\boldsymbol{D}_\ell^p\rangle\right)^{\frac{1}{p}} \leq \left(\sum_{\ell=1}^{L}\alpha_\ell\sum_{ij}[\boldsymbol{D}_\ell]_{ij}^p p_{ij}\sum_{k}\frac{q_{jk}}{\widetilde{b}_j}\right)^{\frac{1}{p}} + \left(\sum_{\ell=1}^{L}\alpha_\ell\sum_{ik}[\boldsymbol{D}_\ell]_{jk}^p q_{jk}\sum_{j}\frac{p_{ij}}{\widetilde{b}_j}\right)^{\frac{1}{p}}
$$

$$
\leq \left(\sum_{\ell=1}^{L}\alpha_\ell\sum_{ij}[\boldsymbol{D}_\ell]_{ij}^p p_{ij}\right)^{\frac{1}{p}} + \left(\sum_{\ell=1}^{L}\alpha_\ell\sum_{ik}[\boldsymbol{D}_\ell]_{jk}^p q_{jk}\right)^{\frac{1}{p}}
$$

$$
\leq \left(\max_{\boldsymbol{\alpha}\in\boldsymbol{\Sigma}^L}\sum_{\ell=1}^{L}\alpha_\ell\sum_{ij}[\boldsymbol{D}_\ell]_{ij}^p p_{ij}\right)^{\frac{1}{p}} + \left(\max_{\boldsymbol{\alpha}\in\boldsymbol{\Sigma}^L}\sum_{\ell=1}^{L}\alpha_\ell\sum_{ik}[\boldsymbol{D}_\ell]_{jk}^p q_{jk}\right)^{\frac{1}{p}}
$$

$$
\leq \text{FRWD}_p(\mu,\nu) + \text{FRWD}_p(\nu,\gamma)
$$

This inequality is valid for all $\alpha$. Therefore, we have

$$
\text{FRWD}_p(\mu,\nu) \leq \text{FRWD}_p(\mu,\nu) + \text{FRWD}_p(\nu,\gamma)
$$

$\square$

FROT WITH LINEAR PROGRAMMING

**Linear Programming:** The FROT is a convex piecewise-linear minimization because the objective is the max of linear functions. Thus, we can solve the FROT problem via linear programming:

$$
\min_{\boldsymbol{\Pi}\in\boldsymbol{U}(\mu,\nu),t} \quad t, \quad \text{s.t.} \quad \langle\boldsymbol{\Pi},\boldsymbol{C}_\ell\rangle \leq t, \ell = 1, 2, \ldots, L.
$$

This optimization can be easily solved using an off-the-shelf LP package. However, the computational cost of this LP problem is high in general (i.e., $O(n^3), n = m$).

The FROT problem can be written as

$$\min_{\mathbf{\Pi}} \quad \max_{\ell \in \{1,2,\ldots,L\}} \langle \mathbf{\Pi}, \mathbf{C}_\ell \rangle,$$
$$\text{s.t.} \quad \mathbf{\Pi}\mathbf{1}_m = \mathbf{a}, \mathbf{\Pi}^\top \mathbf{1}_n = \mathbf{b}, \mathbf{\Pi} \geq 0.$$

This problem can be transformed to an equivalent linear program by first forming an epigraph problem:

$$\min_{\mathbf{\Pi},t} \quad t,$$
$$\text{s.t.} \quad \max_{\ell \in \{1,2,\ldots,L\}} \langle \mathbf{\Pi}, \mathbf{C}_\ell \rangle \leq t$$
$$\mathbf{\Pi}\mathbf{1}_m = \mathbf{a}, \mathbf{\Pi}^\top \mathbf{1}_n = \mathbf{b}, \mathbf{\Pi} \geq 0.$$

Thus, the linear programming for FROT is given as

$$\min_{\mathbf{\Pi},t} \quad t$$
$$\text{s.t.} \quad \langle \mathbf{\Pi}, \mathbf{C}_\ell \rangle \leq t, \ell = 1, 2, \ldots, L$$
$$\mathbf{\Pi}\mathbf{1}_m = \mathbf{a}, \mathbf{\Pi}^\top \mathbf{1}_n = \mathbf{b}, \mathbf{\Pi} \geq 0.$$

Next, we transform this linear programming problem into the canonical form. For matrix $\mathbf{\Pi} = (\boldsymbol{\pi}_1 \ \boldsymbol{\pi}_2 \ \ldots \ \boldsymbol{\pi}_n)^\top \in \mathbb{R}^{n \times m}$ and $\boldsymbol{\pi}_i \in \mathbb{R}^n$, we can vectorize the matrix using the following linewise operator:

$$\text{vec}(\mathbf{\Pi}) = (\boldsymbol{\pi}_1^\top \ \boldsymbol{\pi}_2^\top \ \ldots \ \boldsymbol{\pi}_n^\top)^\top \in \mathbb{R}^{nm}.$$

Using this vectorization operator, we can write $\langle \mathbf{\Pi}, \mathbf{C}_\ell \rangle \leq t$ as

$$\begin{pmatrix} \text{vec}(\mathbf{C}_1)^\top & -1 \\ \text{vec}(\mathbf{C}_2)^\top & -1 \\ \vdots & \vdots \\ \text{vec}(\mathbf{C}_L)^\top & -1 \end{pmatrix} \begin{pmatrix} \text{vec}(\mathbf{\Pi}) \\ t \end{pmatrix} \leq \mathbf{0}_L,$$

where $\mathbf{0}_L \in \mathbb{R}^L$ is a vector whose elements are zero.

For the constraints $\mathbf{\Pi}\mathbf{1}_m = \mathbf{a}$ and $\mathbf{\Pi}^\top \mathbf{1}_n = \mathbf{b}$, we can define vectors $\mathbf{q}_1, \ldots, \mathbf{q}_n \in \mathbb{R}^{nm}$ and $\mathbf{r}_1, \ldots, \mathbf{r}_m \in \mathbb{R}^{nm}$ such that $\mathbf{q}_i^\top \text{vec}(\mathbf{\Pi}) = a_i$ and $\mathbf{r}_j^\top \text{vec}(\mathbf{\Pi}) = b_j$ in this way:

$$\mathbf{q}_1 = (\mathbf{1}_m^\top, \mathbf{0}_m^\top, \ldots, \mathbf{0}_m^\top)^\top,$$
$$\mathbf{q}_2 = (\mathbf{0}_m^\top, \mathbf{1}_m^\top, \ldots, \mathbf{0}_m^\top)^\top,$$
$$\vdots$$
$$\mathbf{q}_n = (\mathbf{0}_m^\top, \mathbf{0}_m^\top, \ldots, \mathbf{1}_m^\top)^\top$$

and

$$\mathbf{r}_1 = (1, \mathbf{0}_{m-1}^\top, 1, \mathbf{0}_{m-1}^\top, \ldots, 1, \mathbf{0}_{m-1}^\top)^\top,$$
$$\mathbf{r}_2 = (0, 1, \mathbf{0}_{m-1}^\top, 1, \mathbf{0}_{m-1}^\top, \ldots, 1, \mathbf{0}_{m-2}^\top)^\top,$$
$$\vdots$$
$$\mathbf{r}_m = (\mathbf{0}_{m-1}^\top, 1, \mathbf{0}_{m-1}^\top, 1, \ldots \mathbf{0}_{m-1}^\top, 1)^\top$$

We can collect these vectors to obtain the vectorized constraints:

$$\begin{pmatrix} \mathbf{q}_1^\top & 0 \\ \mathbf{q}_2^\top & 0 \\ \vdots & \vdots \\ \mathbf{q}_n^\top & 0 \end{pmatrix} \begin{pmatrix} \text{vec}(\mathbf{\Pi}) \\ t \end{pmatrix} = \mathbf{a}, \quad \begin{pmatrix} \mathbf{r}_1^\top & 0 \\ \mathbf{r}_2^\top & 0 \\ \vdots & \vdots \\ \mathbf{r}_m^\top & 0 \end{pmatrix} \begin{pmatrix} \text{vec}(\mathbf{\Pi}) \\ t \end{pmatrix} = \mathbf{b},$$

Thus, we can rewrite the linear programming as

$$\min_{\boldsymbol{u}} \quad \boldsymbol{e}^{\top}\boldsymbol{u}$$

$$\text{s.t.} \quad (\boldsymbol{A}^{\top} - \boldsymbol{1}_L)\boldsymbol{u} \leq \boldsymbol{0}_L, (\boldsymbol{Q}^{\top} \ \boldsymbol{0}_n)\boldsymbol{u} = \boldsymbol{a}, (\boldsymbol{R}^{\top} \ \boldsymbol{0}_m)\boldsymbol{u} = \boldsymbol{b}, \boldsymbol{u} \geq 0,$$

where $\boldsymbol{u} = (\text{vec}(\boldsymbol{\Pi})^{\top} \ t)^{\top} \in \mathbb{R}^{nm+1}$, $\boldsymbol{e} = (\boldsymbol{0}_{nm}^{\top} \ 1)^{\top} \in \mathbb{R}^{nm+1}$ is the unit vector whose $nm + 1$-th element is 1, $\boldsymbol{A} = (\text{vec}(\boldsymbol{C}_1), \dots, \text{vec}(\boldsymbol{C}_L)) \in \mathbb{R}^{nm \times L}$, $\boldsymbol{Q} = (\boldsymbol{q}_1, \dots, \boldsymbol{q}_n) \in \mathbb{R}^{nm \times n}$, and $\boldsymbol{R} = (\boldsymbol{r}_1, \dots, \boldsymbol{r}_m) \in \mathbb{R}^{nm \times m}$. $\boldsymbol{Q} = (\boldsymbol{I}_n \ \boldsymbol{I}_n \ \dots \boldsymbol{I}_n)$

PROOF OF LEMMA 2

We optimize the function with respect to $\boldsymbol{\alpha}$:

$$\max_{\boldsymbol{\alpha}} \quad J(\boldsymbol{\alpha})$$

$$\text{s.t.} \quad \boldsymbol{\alpha}^{\top}\boldsymbol{1}_K = 1, \alpha_1, \dots, \alpha_K \geq 0,$$

where

$$J(\boldsymbol{\alpha}) = \sum_{\ell=1}^{L} \alpha_\ell \phi_\ell - \eta \sum_{\ell=1}^{L} \alpha_\ell(\log \alpha_\ell - 1). \tag{8}$$

Because the entropic regularization is a strong convex function and its negative counterpart is a strong concave function, the maximization problem is a concave optimization problem.

We consider the following objective function with the Lagrange multiplier $\epsilon$:

$$\widetilde{J}(\boldsymbol{\alpha}) = \sum_{\ell=1}^{L} \alpha_\ell \phi_\ell - \eta \sum_{\ell=1}^{L} \alpha_\ell(\log \alpha_\ell - 1) + \epsilon(\boldsymbol{\alpha}^{\top}\boldsymbol{1}_K - 1)$$

Note that owing to the entropic regularization, the nonnegative constraint is automatically satisfied.

Taking the derivative with respect to $\alpha_\ell$, we have

$$\frac{\partial \widetilde{J}(\boldsymbol{\alpha})}{\partial \alpha_\ell} = \phi_\ell - \eta\left(\log \alpha_\ell - 1 + \alpha_\ell \frac{1}{\alpha_\ell}\right) + \epsilon = 0.$$

Thus, the optimal $\alpha_\ell$ has the form

$$\alpha_\ell = \exp\left(\frac{1}{\eta}\phi_\ell\right)\exp\left(\frac{\epsilon}{\eta}\right).$$

$\alpha_\ell$ satisfies the sum to one constraint.

$$\exp\left(\frac{\epsilon}{\eta}\right) = \frac{1}{\sum_{\ell'=1}^{L}\exp\left(\frac{1}{\eta}\phi_{\ell'}\right)}$$

Hence, the optimal $\alpha_\ell$ is given by

$$\alpha_\ell = \frac{\exp\left(\frac{1}{\eta}\phi_\ell\right)}{\sum_{\ell'=1}^{L}\exp\left(\frac{1}{\eta}\phi_{\ell'}\right)}.$$

Substituting this into Eq.(8), we have

$$J(\boldsymbol{\alpha}^*) = \sum_{\ell=1}^{L} \frac{\exp\left(\frac{1}{\eta}\phi_\ell\right)}{\sum_{\ell'=1}^{L}\exp\left(\frac{1}{\eta}\phi_{\ell'}\right)}\phi_\ell - \eta\sum_{\ell=1}^{L}\frac{\exp\left(\frac{1}{\eta}\phi_\ell\right)}{\sum_{\ell'=1}^{L}\exp\left(\frac{1}{\eta}\phi_{\ell'}\right)}\left(\log\left(\frac{\exp\left(\frac{1}{\eta}\phi_\ell\right)}{\sum_{\ell'=1}^{L}\exp\left(\frac{1}{\eta}\phi_{\ell'}\right)}\right) - 1\right)$$

$$= \eta\log\left(\sum_{\ell=1}^{L}\exp\left(\frac{1}{\eta}\phi_\ell\right)\right) + \eta$$

Therefore, the final objective function is given by

$$J(\boldsymbol{\alpha}^*) = \eta \log \left( \sum_{\ell=1}^{L} \exp \left( \frac{1}{\eta} \phi_\ell \right) \right) + \eta$$

$\square$

PROOF OF PROPOSITION 3

Proof: For $0 \leq \theta \leq 1$ and $\eta > 0$, we have

$$\sum_{\ell=1}^{L} \exp \left( \frac{1}{\eta} \langle \theta \boldsymbol{\Pi}_1 + (1-\theta) \boldsymbol{\Pi}_2, \boldsymbol{D}_\ell \rangle \right) = \sum_{\ell=1}^{L} \exp \left( \frac{\theta}{\eta} \langle \boldsymbol{\Pi}_1, \boldsymbol{D}_\ell \rangle + \frac{(1-\theta)}{\eta} \langle \boldsymbol{\Pi}_2, \boldsymbol{D}_\ell \rangle \right)$$

$$= \sum_{\ell=1}^{L} \exp \left( \frac{1}{\eta} \langle \boldsymbol{\Pi}_1, \boldsymbol{D}_\ell \rangle \right)^{\theta} \exp \left( \frac{1}{\eta} \langle \boldsymbol{\Pi}_2, \boldsymbol{D}_\ell \rangle \right)^{1-\theta}$$

$$\leq \left( \sum_{\ell=1}^{L} \exp \left( \frac{1}{\eta} \langle \boldsymbol{\Pi}_1, \boldsymbol{D}_\ell \rangle \right) \right)^{\theta} \left( \sum_{\ell=1}^{L} \exp \left( \frac{1}{\eta} \langle \boldsymbol{\Pi}_2, \boldsymbol{D}_\ell \rangle \right) \right)^{1-\theta}$$

Here, we use Hölder's inequality with $p = 1/\theta$, $q = 1/(1-\theta)$, and $1/p + 1/q = 1$.

Applying a logarithm on both sides of the equation and then premultiplying $\eta$, we have

$$\eta \log \left( \sum_{\ell=1}^{L} \exp \left( \frac{1}{\eta} \langle \theta \boldsymbol{\Pi}_1 + (1-\theta) \boldsymbol{\Pi}_2, \boldsymbol{D}_\ell \rangle \right) \right) \leq \theta \eta \log \left( \sum_{\ell=1}^{L} \exp \left( \frac{1}{\eta} \langle \boldsymbol{\Pi}_1, \boldsymbol{D}_\ell \rangle \right) \right)$$

$$+ (1-\theta) \eta \log \left( \sum_{\ell=1}^{L} \exp \left( \frac{1}{\eta} \langle \boldsymbol{\Pi}_2, \boldsymbol{D}_\ell \rangle \right) \right)$$

$\square$

PROOF OF PROPOSITION 4

**Theorem 5** *(Jaggi, 2013) For each $t \geq 1$, the iterates $\boldsymbol{\Pi}^{(t)}$ of Algorithms 1, 2, 3, and 4 in (Jaggi, 2013) satisfy*

$$f(\boldsymbol{\Pi}^{(t)}) - f(\boldsymbol{\Pi}^*) \leq \frac{2C_f}{t+2}(1+\delta),$$

*where $\boldsymbol{\Pi}^* \in \mathcal{D}$ is an optimal solution to problem*

$$\boldsymbol{\Pi}^* = \underset{\boldsymbol{\Pi} \in \mathcal{D}}{\arg\min} \ f(\boldsymbol{\Pi}),$$

*$C_f$ is the curvature constant defined as*

$$C_f := \sup_{\boldsymbol{\Pi}, \widehat{\boldsymbol{\Pi}}, \gamma} \frac{2}{\gamma^2} (f(\boldsymbol{\Pi}') - f(\boldsymbol{\Pi}) - \langle \boldsymbol{\Pi}' - \boldsymbol{\Pi}, \nabla f(\boldsymbol{\Pi}) \rangle$$

$$\text{s.t. } \boldsymbol{\Pi}, \widehat{\boldsymbol{\Pi}} \in \mathcal{D}, \gamma \in [0, \ 1], \boldsymbol{\Pi}' = \boldsymbol{\Pi} + \gamma(\widehat{\boldsymbol{\Pi}} - \boldsymbol{\Pi}),$$

*$\delta \geq 0$ is the accuracy with which internal linear subproblems are solved.*

**Lemma 6** *(Jaggi, 2013) Let $f$ be a convex and differentiable function with its gradient $\nabla f$ being Lipschitz-continuous w.r.t. some norm $\|\cdot\|$ over the domain $\mathcal{D}$ with Lipschitz-constant $L > 0$. Then,*

$$C_f \leq \text{diam}_{\|\cdot\|}(\mathcal{D})^2 L$$

**Definition 7** *The softmax function is given by*

$$\sigma(\boldsymbol{z}) = \frac{1}{\sum_{\ell'=1}^{L} \exp(\lambda z_{\ell'})} \begin{pmatrix} \exp(\lambda z_1) \\ \exp(\lambda z_2) \\ \vdots \\ \exp(\lambda z_L) \end{pmatrix},$$

*where $\lambda > 0$ is referred to as the inverse temperature constant.*

**Lemma 8** *(Gao & Pavel, 2017) The softmax function $\sigma(\cdot)$ is L-Lipschitz with respect to $\|\cdot\|_2$ with $L = \lambda$, that is for all $\boldsymbol{z}, \boldsymbol{z}' \in \mathbb{R}^n$,*

$$\|\sigma(\boldsymbol{z}) - \sigma(\boldsymbol{z}')\|_2 \le \lambda \|\boldsymbol{z} - \boldsymbol{z}'\|_2,$$

*where $\lambda$ is the inverse temperature constant.*

The derivative of $G_\eta(\boldsymbol{\Pi})$ is given as

$$\frac{\partial G_\eta(\boldsymbol{\Pi})}{\partial \boldsymbol{\Pi}} = \sum_{\ell=1}^{L} \frac{\exp\left(\frac{1}{\eta}\langle \boldsymbol{\Pi}, \boldsymbol{C}_\ell \rangle\right)}{\sum_{\ell'=1}^{L} \exp\left(\frac{1}{\eta}\langle \boldsymbol{\Pi}, \boldsymbol{C}_{\ell'} \rangle\right)} \boldsymbol{C}_\ell = \boldsymbol{M}_{\boldsymbol{\Pi}}.$$

Thus, we have

$$\text{vec}(\boldsymbol{M}_{\boldsymbol{\Pi}}) = \boldsymbol{\Phi} \boldsymbol{p}_{\boldsymbol{\Pi}},$$

where

$$\boldsymbol{\Phi} = (\text{vec}(\boldsymbol{C}_1), \text{vec}(\boldsymbol{C}_2), \dots, \text{vec}(\boldsymbol{C}_L)) \in \mathbb{R}^{nm \times L},$$

$$\boldsymbol{p}_{\boldsymbol{\Pi}} = \left( \frac{\exp\left(\frac{1}{\eta}\langle \boldsymbol{\Pi}, \boldsymbol{C}_1 \rangle\right)}{\sum_{\ell'=1}^{L} \exp\left(\frac{1}{\eta}\langle \boldsymbol{\Pi}, \boldsymbol{C}_{\ell'} \rangle\right)}, \dots, \frac{\exp\left(\frac{1}{\eta}\langle \boldsymbol{\Pi}, \boldsymbol{C}_L \rangle\right)}{\sum_{\ell'=1}^{L} \exp\left(\frac{1}{\eta}\langle \boldsymbol{\Pi}, \boldsymbol{C}_{\ell'} \rangle\right)} \right)^\top \in \mathbb{R}^L.$$

Here, $\boldsymbol{p}_{\boldsymbol{\Pi}}$ is the softmax function $\sigma(\boldsymbol{z})$ with $z_\ell = \langle \boldsymbol{\Pi}, \boldsymbol{C}_\ell \rangle$.

We have

$$\begin{aligned} \|\nabla G_\eta(\boldsymbol{\Pi}) - \nabla G_\eta(\boldsymbol{\Pi}')\|_2 &= \|\boldsymbol{\Phi} \boldsymbol{p}_{\boldsymbol{\Pi}} - \boldsymbol{\Phi} \boldsymbol{p}_{\boldsymbol{\Pi}'}\|_2 \\ &\le \|\boldsymbol{\Phi}\|_{\text{op}} \|\boldsymbol{p}_{\boldsymbol{\Pi}} - \boldsymbol{p}_{\boldsymbol{\Pi}'}\|_2, \\ &\le \frac{1}{\eta} \|\boldsymbol{\Phi}\|_{\text{op}} \|\boldsymbol{\Phi}^\top \text{vec}(\boldsymbol{\Pi}) - \boldsymbol{\Phi}^\top \text{vec}(\boldsymbol{\Pi}')\|_2 \quad (\text{Lemma 8 with } \lambda = \frac{1}{\eta}) \\ &\le \frac{1}{\eta} \|\boldsymbol{\Phi}\|_{\text{op}} \|\boldsymbol{\Phi}^\top\|_{\text{op}} \|\text{vec}(\boldsymbol{\Pi}) - \text{vec}(\boldsymbol{\Pi}')\|_2 \end{aligned}$$

where $\|\cdot\|_{\text{op}}$ is the operator norm. We have $\|\boldsymbol{\Phi}\|_{\text{op}} = \|\boldsymbol{\Phi}^\top\|_{\text{op}}$, $\|\boldsymbol{\Phi}^\top \boldsymbol{\Phi}\|_{\text{op}} = \|\boldsymbol{\Phi}\|_{\text{op}}^2$, and $\|\text{vec}(\boldsymbol{\Pi}) - \text{vec}(\boldsymbol{\Pi}')\|_2 \le \sqrt{2}$. Therefore, the Lipschitz constant for the gradient is $L = \frac{1}{\eta}\|\boldsymbol{\Phi}\|_{\text{op}}^2 = \frac{1}{\eta}\sigma_{\max}(\boldsymbol{\Phi}^\top \boldsymbol{\Phi})$, and the curvature constant is bounded above by $C_f \le 2L$, where $\sigma_{max}(\boldsymbol{\Phi}^\top \boldsymbol{\Phi})$ is the largest eigenvalue of the matrix $\boldsymbol{\Phi}^\top \boldsymbol{\Phi}$. By plugging $C_f$ in Theorem 5, we have

$$G_\eta(\boldsymbol{\Pi}^{(t)}) - G_\eta(\boldsymbol{\Pi}^*) \le \frac{4\sigma_{max}(\boldsymbol{\Phi}^\top \boldsymbol{\Phi})}{\eta(t+2)}(1 + \delta).$$

MAX/MIN FORMULATION

We define the max–min formulation of the FROT as

$$\max_{\boldsymbol{\alpha} \in \boldsymbol{\Sigma}^L} \sum_{\ell=1}^{L} \alpha_\ell \min_{\boldsymbol{\Pi} \in U(\boldsymbol{a}_\ell, \boldsymbol{b}_\ell)} \sum_{i=1}^{n} \sum_{j=1}^{m} \pi_{ij} c(\boldsymbol{x}_i^{(\ell)}, \boldsymbol{y}_j^{(\ell)}),$$

where $\boldsymbol{\Sigma}^L = \{\boldsymbol{\alpha} \in \mathbb{R}_+^L : \boldsymbol{\alpha}^\top \mathbf{1}_L = 1\}$ is the probability simplex, the set of probability vectors in $\mathbb{R}^L$.

This problem can be solved by computing the group that maximizes the optimal transport distance $k^* = \mathrm{argmax}_k W_1(\mu^{(\ell)}, \nu^{(\ell)})$ and then by considering $\alpha^* = \delta_{k^*}$ as a one-hot vector.

The result of this formulation provides an intuitive idea (the same as for the robust Wasserstein method). Hence, we maximize the group (instead of the subspace) that provides the optimal result. However, the formulation requires solving the OT problem $L$ times. This approach may not be suitable if we have a large $L$. Moreover, the argmax function is generally not differentiable.

**Relation to the max-sliced Wasserstein distance:** The max-sliced Wasserstein-2 distance can be defined as (Deshpande et al., 2019)

$$\text{max-}W_2(\mu, \nu) = \left( \max_{\boldsymbol{w} \in \boldsymbol{\Omega}} \quad \min_{\boldsymbol{\Pi} \in \boldsymbol{U}(\boldsymbol{a}_\ell, \boldsymbol{b}_\ell)} \sum_{i=1}^{n} \sum_{j=1}^{m} \pi_{ij} (\boldsymbol{w}^\top \boldsymbol{x}_i - \boldsymbol{w}^\top \boldsymbol{y}_j)^2 \right)^{\frac{1}{2}},$$

where $\boldsymbol{\Omega} \subset \mathbb{R}^d$ is the set of all possible directions on the unit sphere.

The max–sliced Wasserstein is a *max–min* approach. That is, for each $\boldsymbol{w}$, it requires solving the OT problem. The max–min approach is suited for simply measuring the divergence between two distributions. However, it is difficult to interpret features using the max–sliced Wasserstein, where it is the key motivation of FROT.

**Relation to Subspace Robust Wasserstein (Paty & Cuturi, 2019):** Here, we show that 2-FRWD with $d(\boldsymbol{x}, \boldsymbol{y}) = \|\boldsymbol{x} - \boldsymbol{y}\|_2$ is a special case of SRW. Let us define $\boldsymbol{U} = (\sqrt{\alpha_1}\boldsymbol{e}_1, \sqrt{\alpha_2}\boldsymbol{e}_2, \ldots, \sqrt{\alpha_d}\boldsymbol{e}_d)^\top \in \mathbb{R}^{d \times d}$, where $\boldsymbol{e}_\ell \in \mathbb{R}^d$ is the one-hot vector whose $\ell$th element is 1 and $\boldsymbol{\alpha}^\top \mathbf{1} = 1, \alpha_\ell \geq 0$. Then, the objective function of SRW can be written as

$$\sum_{i=1}^{n} \sum_{j=1}^{m} \pi_{ij} \|\boldsymbol{U}^\top \boldsymbol{x}_i - \boldsymbol{U}^\top \boldsymbol{y}_j\|_2^2 = \sum_{i=1}^{n} \sum_{j=1}^{m} \pi_{ij} (\boldsymbol{x}_i - \boldsymbol{y}_j)^\top \boldsymbol{U}\boldsymbol{U}^\top (\boldsymbol{x}_i - \boldsymbol{y}_j)$$

$$= \sum_{i=1}^{n} \sum_{j=1}^{m} \pi_{ij} (\boldsymbol{x}_i - \boldsymbol{y}_j)^\top \mathrm{diag}(\boldsymbol{\alpha}) (\boldsymbol{x}_i - \boldsymbol{y}_j)$$

$$= \sum_{i=1}^{n} \sum_{j=1}^{m} \pi_{ij} \sum_{\ell=1}^{d} \alpha_\ell (x_i^{(\ell)} - y_j^{(\ell)})^2.$$

Therefore, SRW and 2-FRWD are equivalent if we set $\boldsymbol{U} = (\sqrt{\alpha_1}\boldsymbol{e}_1, \sqrt{\alpha_2}\boldsymbol{e}_2, \ldots, \sqrt{\alpha_d}\boldsymbol{e}_d)^\top$ and $d(\boldsymbol{x}, \boldsymbol{y}) = \|\boldsymbol{x} - \boldsymbol{y}\|_2$.

Table 2: Computational time comparison (s) for feature selection from biological datasets.

| Data | $d$ | $n$ | Wasserstein (Sort) | Linear | MMD | FROT |
|------|-----|-----|--------------------|--------|-----|------|
| Colon | 2000 | 62 | 12.57 ($\pm$ 3.27) | 0.00 ($\pm$ 0.00) | 1.36 ($\pm$ 0.15) | 0.41 ($\pm$ 0.07) |
| Leukemia | 7070 | 72 | 46.76 ($\pm$ 19.47) | 0.01 ($\pm$ 0.00) | 5.03 ($\pm$ 0.79) | 1.13 ($\pm$ 0.14) |
| Prostate_GE | 5966 | 102 | 51.99 ($\pm$ 16.37) | 0.02 ($\pm$ 0.00) | 6.01 ($\pm$ 1.17) | 1.04 ($\pm$ 0.11) |
| GLI_85 | 22283 | 85 | 142.1 ($\pm$ 21.65) | 0.04 ($\pm$ 0.00) | 23.6 ($\pm$ 1.21) | 3.44 ($\pm$ 0.36) |

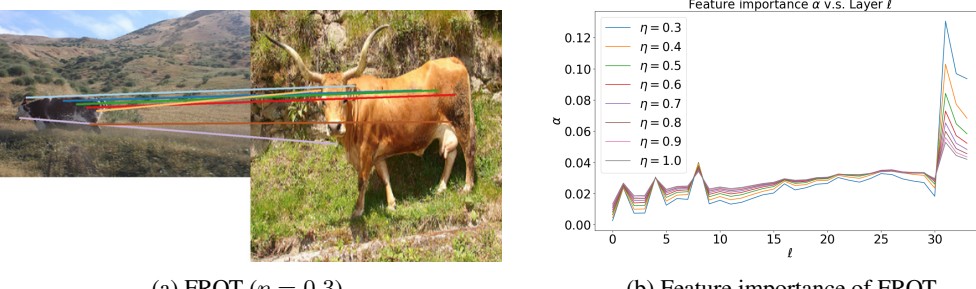

(a) FROT ($\eta = 0.3$).    (b) Feature importance of FROT.

Figure 5: One-to-one matching results of FROT ($\eta = 0.3$) and feature importance of FROT.

ADDITIONAL SEMANTIC CORRESPONDENCE EXPERIMENTS

Figure 5a shows an example of key points matched using the FROT algorithm. Fig.5b shows the corresponding feature importance. The lower the $\eta$ value, the smaller the number of layers used. The interesting finding here is that the selected important layer in this case is the third layer from the last. More qualitative results are presented in the Figure 7.

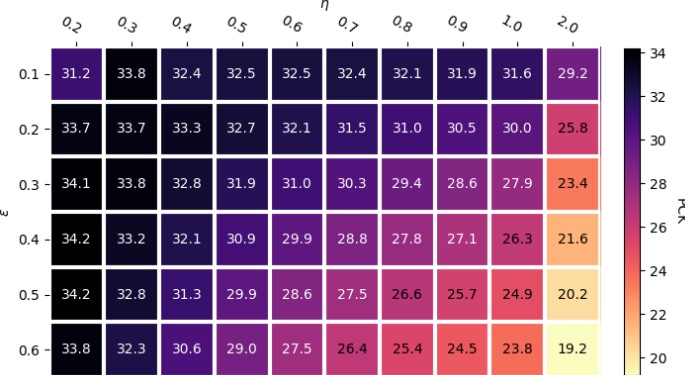

Figure 6: Average PCK scores on the validation split of SPair-71k with different pairs of hyperparameters.

Table 3: Per-class PCK ($\alpha_{bbox} = 0.1$) results using the SPair-71k. All models use ResNet101 as the backbone.

| | Methods | aero | bike | bird | boat | bottle | bus | car | cat | chair | cow | dog | horse | moto | person | plant | sheep | train | tv | all |
|---|---|---|---|---|---|---|---|---|---|---|---|---|---|---|---|---|---|---|---|---|
| Authors' original models | CNNGeo (Rocco et al., 2017) | 21.3 | 15.1 | 34.6 | 12.8 | 31.2 | 26.3 | 24.0 | 30.6 | 11.6 | 24.3 | 20.4 | 12.2 | 19.7 | 15.6 | 14.3 | 9.6 | 28.5 | 28.8 | 18.1 |
| | A2Net (Hongsuck Seo et al., 2018) | 20.8 | 17.1 | 37.4 | 13.9 | 33.6 | 29.4 | 26.5 | 34.9 | 12.0 | 26.5 | 22.5 | 13.3 | 21.3 | 20.0 | 16.9 | 11.5 | 28.9 | 31.6 | 20.1 |
| | WeakAlign (Rocco et al., 2018a) | 23.4 | 17.0 | 41.6 | 14.6 | 37.6 | 28.1 | 26.6 | 32.6 | 12.6 | 27.9 | 23.0 | 13.6 | 21.3 | 22.2 | 17.9 | 10.9 | 31.5 | 34.8 | 21.1 |
| | NC-Net (Rocco et al., 2018b) | 24.0 | 16.0 | 45.0 | 13.7 | 35.7 | 25.9 | 19.0 | 50.4 | 14.3 | 32.6 | 27.4 | 19.2 | 21.7 | 20.3 | 20.4 | 13.6 | 33.6 | 40.4 | 26.4 |
| SPair-71k finetuned models | CNNGeo | 23.4 | 16.7 | 40.2 | 14.3 | 36.4 | 27.7 | 26.0 | 32.7 | 12.7 | 27.4 | 22.8 | 13.7 | 20.9 | 21.0 | 17.5 | 10.2 | 30.8 | 34.1 | 20.6 |
| | A2Net | 22.6 | 18.5 | 42.0 | 16.4 | 37.9 | **30.8** | 26.5 | 35.6 | 13.3 | 29.6 | 24.3 | 16.0 | 21.6 | 22.8 | 20.5 | 13.5 | 31.4 | 36.5 | 22.3 |
| | WeakAlign | 22.2 | 17.6 | 41.9 | 15.1 | 38.1 | 27.4 | **27.2** | 31.8 | 12.8 | 26.8 | 22.6 | 14.2 | 20.0 | 22.2 | 17.9 | 10.4 | 32.2 | 35.1 | 20.9 |
| | NC-Net | 17.9 | 12.2 | 32.1 | 11.7 | 29.0 | 19.9 | 16.1 | 39.2 | 9.9 | 23.9 | 18.8 | 15.7 | 17.4 | 15.9 | 14.8 | 9.6 | 24.2 | 31.1 | 20.1 |
| SPair-71k validation | HPF | 25.2 | 18.9 | 52.1 | 15.7 | 38.0 | 22.8 | 19.1 | 52.9 | 17.9 | 33.0 | 32.8 | 20.6 | 24.4 | 27.9 | 21.1 | 15.9 | 31.5 | 35.6 | 28.2 |
| | OT-HPF | 32.6 | 18.9 | **62.5** | 20.7 | **42.0** | 26.1 | 20.4 | 61.4 | **19.7** | **41.3** | **41.7** | **29.8** | **29.6** | **31.8** | 25.0 | **23.5** | 44.7 | 37.0 | 33.9 |
| Without SPair-71k validation | OT | 30.1 | 16.5 | 50.4 | 17.3 | 38.0 | 22.9 | 19.7 | 54.3 | 17.0 | 28.4 | 31.3 | 22.1 | 28.0 | 19.5 | 21.0 | 17.8 | 42.6 | 28.8 | 28.3 |
| | FROT ($\eta = 0.2$) | 34.0 | 17.2 | 55.6 | 19.7 | 39.6 | 24.3 | 19.9 | 57.9 | 15.8 | 33.1 | 34.0 | 24.8 | 26.1 | 28.5 | 23.1 | 21.2 | 43.4 | 33.6 | 30.8 |
| | FROT ($\eta = 0.3$) | **35.0** | **20.9** | 56.3 | **23.4** | 40.7 | 27.2 | 21.9 | 62.0 | 17.5 | 38.8 | 36.2 | 27.9 | 28.0 | 30.4 | **26.9** | 23.1 | 49.7 | 38.4 | 33.7 |
| | FROT ($\eta = 0.4$) | 34.0 | 18.7 | 57.0 | 20.0 | 39.9 | 25.9 | 19.7 | 61.6 | 17.2 | 38.1 | 36.6 | 26.5 | 26.6 | 27.4 | 26.8 | 22.6 | 49.8 | 38.4 | 32.8 |
| | FROT ($\eta = 0.5$) | 34.1 | 18.8 | 56.9 | 19.9 | 40.0 | 25.6 | 19.2 | 61.9 | 17.4 | 38.7 | 36.5 | 25.6 | 26.9 | 27.2 | 26.3 | 22.1 | **50.3** | 38.6 | 32.8 |
| | FROT ($\eta = 0.6$) | 33.8 | 19.3 | 56.5 | 19.9 | 39.9 | 25.9 | 19.2 | 62.3 | 17.7 | 38.4 | 36.6 | 26.0 | 27.2 | 27.0 | 26.1 | 22.2 | 50.1 | **39.2** | 32.8 |
| | FROT ($\eta = 0.7$) | 33.4 | 19.4 | 56.6 | 20.0 | 39.6 | 26.1 | 19.1 | **62.4** | 17.9 | 38.0 | 36.5 | 26.0 | 27.5 | 26.5 | 25.5 | 21.6 | 49.7 | 38.9 | 32.7 |
| | FROT ($\eta = 0.8$) | 33.2 | 19.0 | 56.2 | 19.8 | 39.4 | 26.2 | 19.6 | 62.3 | 17.3 | 37.5 | 36.5 | 25.8 | 26.5 | 26.0 | 25.2 | 21.3 | 48.9 | 38.2 | 32.3 |
| | FROT ($\eta = 0.9$) | 32.9 | 19.1 | 56.0 | 19.6 | 39.3 | 26.1 | 19.8 | 61.9 | 17.2 | 37.1 | 36.4 | 25.5 | 27.0 | 25.3 | 24.8 | 21.3 | 48.2 | 37.8 | 32.1 |
| | FROT ($\eta = 1.0$) | 32.8 | 19.1 | 55.8 | 19.8 | 39.1 | 25.7 | 19.7 | 61.5 | 17.2 | 37.1 | 35.9 | 25.1 | 27.2 | 25.0 | 24.7 | 21.4 | 47.7 | 37.8 | 32.0 |
| | FROT ($\eta = 2.0$) | 30.0 | 17.5 | 54.6 | 18.2 | 36.6 | 24.3 | 18.9 | 57.7 | 16.8 | 33.8 | 34.7 | 23.1 | 25.8 | 21.1 | 21.5 | 19.5 | 41.6 | 34.0 | 29.5 |
| | FROT ($\eta = 3.0$) | 28.8 | 16.2 | 53.0 | 17.1 | 34.7 | 23.0 | 18.3 | 54.5 | 15.7 | 31.0 | 32.4 | 21.5 | 24.3 | 17.9 | 19.4 | 18.2 | 37.1 | 30.7 | 27.5 |
| | FROT ($\eta = 4.0$) | 27.8 | 15.2 | 52.0 | 16.4 | 33.7 | 21.6 | 17.0 | 51.0 | 15.8 | 28.9 | 30.9 | 20.3 | 22.7 | 16.3 | 18.4 | 16.9 | 34.0 | 28.2 | 26.1 |
| | FROT ($\eta = 5.0$) | 26.9 | 14.9 | 50.8 | 16.0 | 32.4 | 20.5 | 16.3 | 48.4 | 15.0 | 27.0 | 30.2 | 19.4 | 21.0 | 14.8 | 17.5 | 15.9 | 32.1 | 26.7 | 24.9 |
| | SRW ($k = 10, \epsilon = 0.1, T = 10$, layer=34) | 23.1 | 9.6 | 26.3 | 12.8 | 27.3 | 15.4 | 12.9 | 29.9 | 11.6 | 16.0 | 13.8 | 12.4 | 19.2 | 8.5 | 11.7 | 10.0 | 31.1 | 12.7 | 17.2 |
| | SRW ($k = 20, \epsilon = 0.1$, layer=34) | 24.1 | 10.7 | 28.6 | 12.9 | 27.8 | 16.9 | 13.7 | 34.7 | 11.1 | 17.1 | 15.9 | 13.4 | 19.7 | 9.7 | 12.2 | 10.3 | 32.5 | 14.4 | 18.3 |
| | SRW ($k = 30, \epsilon = 0.1, T = 10$, layer=34) | 24.4 | 11.2 | 29.8 | 13.2 | 28.3 | 16.7 | 14.1 | 37.1 | 11.5 | 17.3 | 16.2 | 13.9 | 21.1 | 9.8 | 12.9 | 11.7 | 32.6 | 14.7 | 18.9 |
| | SRW ($k = 40, \epsilon = 0.1, T = 10$, layer=34) | 25.0 | 11.5 | 31.0 | 13.3 | 27.9 | 16.6 | 14.1 | 37.5 | 11.4 | 17.4 | 16.8 | 14.5 | 21.5 | 10.0 | 13.1 | 11.2 | 33.0 | 14.9 | 19.1 |
| | SRW ($k = 50, \epsilon = 0.1, T = 10$, layer=34) | 25.3 | 11.4 | 31.2 | 12.9 | 28.0 | 17.2 | 14.8 | 38.0 | 11.4 | 17.4 | 16.9 | 14.8 | 21.7 | 10.4 | 12.9 | 11.6 | 33.2 | 15.0 | 19.3 |
| | SRW ($k = 60, \epsilon = 0.1, T = 10$, layer=34) | 25.3 | 11.6 | 31.5 | 13.1 | 28.0 | 17.2 | 14.8 | 38.3 | 11.3 | 17.5 | 17.4 | 14.5 | 21.8 | 10.5 | 13.3 | 11.4 | 32.9 | 15.0 | 19.4 |
| | SRW ($k = 70, \epsilon = 0.1, T = 10$, layer=34) | 25.2 | 11.7 | 31.3 | 13.1 | 27.8 | 17.3 | 14.8 | 38.4 | 11.4 | 17.6 | 17.0 | 14.6 | 21.6 | 10.4 | 13.1 | 11.5 | 33.0 | 14.9 | 19.3 |
| | SRW ($k = 80, \epsilon = 0.1, T = 10$, layer=34) | 25.1 | 11.7 | 31.2 | 13.0 | 27.8 | 17.2 | 14.8 | 38.5 | 11.5 | 17.5 | 16.9 | 14.7 | 21.9 | 10.3 | 13.0 | 11.4 | 32.9 | 14.9 | 19.3 |
| | SRW ($k = 90, \epsilon = 0.1, T = 10$, layer=34) | 25.2 | 11.5 | 31.2 | 13.2 | 27.8 | 17.4 | 14.9 | 38.4 | 11.5 | 17.4 | 16.9 | 14.7 | 21.8 | 10.4 | 12.6 | 11.5 | 32.9 | 15.0 | 19.3 |
| | SRW ($k = 100, \epsilon = 0.1, T = 10$, layer=34) | 25.2 | 11.5 | 31.1 | 13.0 | 27.8 | 17.4 | 14.8 | 38.4 | 11.5 | 17.4 | 16.8 | 14.6 | 21.8 | 10.7 | 12.9 | 11.4 | 33.0 | 14.9 | 19.3 |
| | SRW ($k = 50, \epsilon = 0.1, T = 3$, layers = {1, 32–34}) | 29.4 | 14.0 | 43.7 | 15.6 | 33.8 | 21.0 | 17.6 | 48.0 | 12.9 | 23.3 | 26.5 | 19.8 | 25.5 | 17.6 | 16.7 | 15.2 | 37.1 | 20.5 | 24.5 |
| | SRW ($k = 50, \epsilon = 0.1, T = 3$, layers = {1, 31–34}) | 29.7 | 14.3 | 44.3 | 15.7 | 34.2 | 21.3 | 17.8 | 48.5 | 13.1 | 23.6 | 27.1 | 20.0 | 25.8 | 18.1 | 16.9 | 15.2 | 37.3 | 21.0 | 24.8 |
| | SRW ($k = 50, \epsilon = 0.1, T = 3$, layers = {1, 30–34}) | 29.8 | 14.7 | 45.6 | 15.9 | 34.8 | 21.5 | 18.0 | 49.3 | 13.3 | 24.0 | 27.7 | 20.6 | 25.7 | 18.7 | 17.2 | 15.3 | 37.7 | 21.5 | 25.2 |

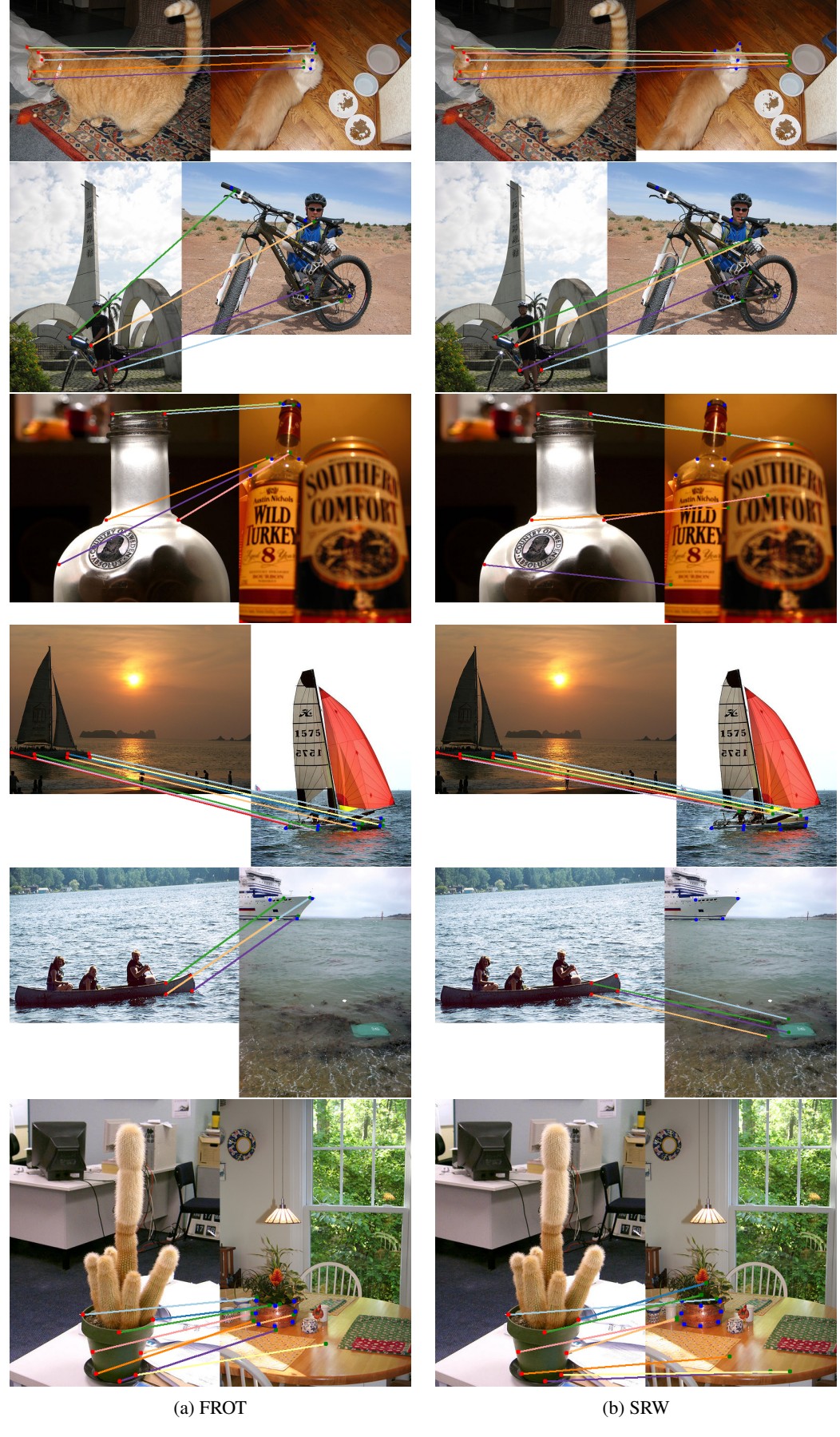

(a) FROT            (b) SRW

Figure 7: Qualitative examples sampled from SPair-71k. For each image, the red points represent the keypoints to be matched while the green points denote the prediction and the blue ones denote the ground truth.

