# OpenReview forum: "Feature-Robust Optimal Transport for High-Dimensional Data"
_ICLR.cc/2021/Conference — Reject_

### Official Review · AnonReviewer4 · 2020-10-24
**FROT is interesting but the analysis is suspicious**

**Rating:** 3
**Confidence:** 3

**Review:**

*Summary*:
The authors try to solve a special kind of high-dimensional optimal transport problem. Specifically, they consider the cases when features are grouped and the grouping is known a-priori. The authors formulate the problem into the feature-robust optimal transport (FROT) problem.
The authors propose two solving algorithms, one based on the Frank-Wolfe method, and one based on linear programming.

*Pros*:
The connection to the feature group sounds interesting to me, as it has a natural connection to the structure of deep learning models.
The presentation (other than the introduction) is easy to follow.

*Cons*:
Note that the first point is the main contributing factor for my rating.

1. Section 3.1 is very confusing, and it seems to me that the authors fail to establish the correct convergence guarantee.
As in page 4, the target is $min_{\pi} max_{\alpha} J(\Pi, \alpha)$
If we fix $\pi$, we can solve for the optimal $\alpha$. Plug this optimal $\alpha$ back in and we obtain $G(\Pi)$.
Intuitively one may choose to solve for $\alpha$ and $\pi$ alternatingly.
However the convergence of $G(\Pi)$ says nothing more than, in a fixed iteration, one can solve exactly for the optimal $\alpha$ and up to $\epsilon$ accuracy for $\Pi$.
We still don't know if the solution of the algorithm indeed minimizes the said loss.
I checked the proof of proposition 4. It just invokes the standard FW-convergence analysis from Jaggi 2013, and argue nothing about the alternative part. Note that, even though the two subproblems (for $\alpha$ and for $\pi$) can be solved almost exactly, it could be non-trivial to set up the convergence of the entire alternating algorithm.
Alternatively, maybe the authors want to argue that solving $min_{\pi} max_{\alpha} J(\Pi, \alpha)$ is equivalent to solving $\max_{\pi} G(\Pi)$. However, this is also not obviously true for me.

2.  What are the other potential applications of FROT? While Semantic Correpondance is an interesting application, I find it hard to convince myself that FROT is better than Liu's 2020-CVPR work (requiring validation dataset is not a big problem - you can always to train-val split). With its similarity to group lasso, FROT might have more interesting applications.

3. Presentation of the introduction can be improved. I find it hard to parse the introduction until I almost finished reading the entire paper. Putting figure 1 to page 2 only creates more questions in my head instead of offering intuitions. Also, it would be helpful if the author can list their contributions in a more organized way.

4. I didn't quite get the high dimensional part. While 'high-dimensional' appears in the abstract, introduction, and conclusion section, I didn't find the correspondence in the main text.

5. I didn't get the robust part, other than the empirical performance in the evaluation section.

---

> ### Author Response · Authors · 2020-11-12
> **We would like to correct your misunderstanding.**
>
> (Improved the readability of reply)
>
> Thank you for your time to review our paper. We will revise based on your comments. For now, we would like to correct your misunderstandings based on the current submission.
>
> **Section 3.1 is very confusing, and it seems to me that the authors fail to establish the correct convergence guarantee.**
>
> We would like to correct your misunderstanding. First, our optimization methods (Frank-Wolfe and Linear programming)  is not a two-step approach. More specifically, we optimize Eq. (5) for Frank-Wolfe and it does not include alpha parameter optimization thanks to Lemma 2. Based on Proposition 3, we can show Eq. (5) is convex with respect to \Pi, and thus we can directly use the Frank-Wolfe algorithm and its analysis (Frank & Wolfe, 1956; Jaggi, 2013),
>
> **What are the other potential applications of FROT? While Semantic Correpondance is an interesting application, I find it hard to convince myself that FROT is better than Liu's 2020-CVPR work (requiring validation dataset is not a big problem - you can always to train-val split). With its similarity to group lasso, FROT might have more interesting applications.**
>
> Thank you for the suggestion. We included the FROT using validation (Table 1 FROT(\eta= 0.2,  \epsilon= 0.4)). Using the validation set, we could get \eta = 0.2 and \epsilon = 0.4, respectively. For the semantic correspondence experiments, FROT got the state-of-the-art performance (34.7), while the current state-of-the-art is Liu’s 2020-CVPR work (33.9).
>
> Moreover, due to the space limitation, we could not include the feature selection experiments in the main paper (it is in the supplementary material). For feature selection tasks, FROT compares favorably with Wasserstein distance-based feature selection algorithm in performance. Moreover, FROT is about two orders of magnitude faster than the Wasserstein distance and is also faster than Maximum Mean Discrepancy (MMD). We will include the feature selection experiments in the main paper.
>
> We are considering using our method of biological data because the group lasso is heavily used in the biological domain. We believe this is an interesting direction.
>
> **Presentation of the introduction can be improved. I find it hard to parse the introduction until I almost finished reading the entire paper. Putting figure 1 to page 2 only creates more questions in my head instead of offering intuitions. Also, it would be helpful if the author can list their contributions in a more organized way.**
>
> Thank you for your suggestion. We will include the contribution statement in the revised manuscript as follows.
>
>  - We propose a feature robust optimal transport (FROT) problem and derive a simple and efficient Frank--Wolfe based algorithm. Furthermore, we propose a feature-robust Wasserstein distance (FRWD).
> - We apply FROT to a high-dimensional feature selection problem and show that FROT is consistent with the Wasserstein distance-based feature selection algorithm with less computational cost than the original algorithm.
> - We used FROT for the layer selection problem in a semantic correspondence problem and showed that the proposed algorithm outperforms existing baseline algorithms.
>
> **I didn't quite get the high dimensional part. While 'high-dimensional' appears in the abstract, introduction, and conclusion section, I didn't find the correspondence in the main text.**
>
> The semantic correspondence data is indeed high-dimensional data. The dimension is d = 32, 576, while the number of samples for each image is a couple of hundred (i.e., d >> n).  Also, in the feature selection experiments in the supplementary material, we used high-dimensional data (see Table 2).
>
> **I didn't get the robust part, other than the empirical performance in the evaluation section.**
>
> In Figure 1, we evaluated FROT and the OT (Cuturi 2013). FROT can get very similar transport plans (Figure 1(c))  to the one obtained by OT from clean data (Figure 1(a)), while OT with noisy data (Figure 1(b)) is different from the OT with clean data (Figure 1(a)).

---

### Official Review · AnonReviewer3 · 2020-10-26
**FROT -- feature-robust optimal transport**

**Rating:** 4
**Confidence:** 4

**Review:**

The proposed framework FROT - feature-robust optimal transport - seeks to select feature groups to both speed up OT computation for high-dimensional data and make it more robust to noise. The exposition is generally clear. My main concerns are limited novelty and lack of extensive experiments.

The paper draws the contrast between prior work SRW that yields a discriminative subspace via dimensionality reduction, and offers a dual perspective to use feature selection instead. A thorough discussion on the pros and cons of feature selection vs feature dimensionality reduction would add insight.

Traditional entropy-regularized OT regularizes using the entropy of the transport plan $\Pi$, whereas FROT regularizes using the probability distribution $\alpha$. One expects a discussion on the effect of this choice.

Currently, the optimization for the group selection is done independently of optimization that produces the features, for instance by the choice of a pretrained network in the semantic correspondence application. It's worthwhile to explore joint optimization of feature generation and selection for downstream tasks.

For the claim of robustness to noise, experiments on data of dimension higher than 10 would be desirable.

There should be more extensive experiments applying FROT to more tasks and compared with additional baselines. Figure 3 compares the objective scores of FW-EMD, FW-Sinkhorn with that of exact OT, there are many other OT algorithms, such as tree-based methods, as referenced in the paper, that can be compared with. These plots should also include variations of the metric across trials. Similarly for the semantic correspondence results in Table 1.

Thank you authors for your response.

---

> ### Author Response · Authors · 2020-11-13
> **Thank you for the review.**
>
> Thank you for your feedback.
>
> **Traditional entropy-regularized OT regularizes using the entropy of the transport plan Π, whereas FROT regularizes using the probability distribution α. One expects a discussion on the effect of this choice.**
>
> In our paper, we regularize both \Pi and \alpha. Specifically, if we regularize \alpha and solve the optimization problem for \alpha, we can get Eq. (5). To optimize Eq. (5), we can update the transport plan by solving linear programming (EMD) or Sinkhorn algorithm (regularized version).
>
> **Currently, the optimization for the group selection is done independently of optimization that produces the features, for instance by the choice of a pretrained network in the semantic correspondence application. It's worthwhile to explore joint optimization of feature generation and selection for downstream tasks.**
>
> Thank you for your valuable feedback. Learning the network parameter would be an interesting idea. In this paper, we focus on proposing a new optimal transport method. And, the joint training of OT and DNN models is actually a new method and we would like to work on this direction as future work.
>
> **For the claim of robustness to noise, experiments on data of dimension higher than 10 would be desirable.**
>
> The dimension of the semantic correspondence dataset is d = 32, 576 as described in the submitted paper. We will add more synthetic experiments.
>
> **There should be more extensive experiments applying FROT to more tasks and compared with additional baselines. Figure 3 compares the objective scores of FW-EMD, FW-Sinkhorn with that of exact OT, there are many other OT algorithms, such as tree-based methods, as referenced in the paper, that can be compared with. These plots should also include variations of the metric across trials. Similarly for the semantic correspondence results in Table 1.**
>
> We would like to emphasize that the Sinkhorn algorithm (Liu CVPR 2020) is a state-of-the-art method for the semantic corresponding task, and the SRW variant is also not reported anywhere.  Moreover, to our knowledge, the state-of-the-art Robust OT method is SRW and could not find any Robust counterparts at the time of submission. Note that the tree-Wasserstein is a method for computing the Wasserstein distance on tree, and it can compute the Wasserstein distance in linear time. However, it is not a robust OT method and the performance of tree-Wasserstein is in general comparable to OT with Sinkhorn algorithm. Of course, if the main contribution is the computational time, we should compare FROT with the tree-Wasserstein. However, this is out of the scope of our paper.

---

### Official Review · AnonReviewer2 · 2020-10-28
**Seems a bit incremental in terms of novelty and contribution, but well-written overall.**

**Rating:** 6
**Confidence:** 4

**Review:**

This work proposes variants of robust OT/p-wasserstein-dist (3)/(4), where the ground cost is in some sense the maximum over costs with (prefixed) groups of features. The motivation is similar to that for feature selection: where perhaps only few of these groups of features are critical/sufficient for OT purposes. So it can also be understood as joint feature-group selection with OT. The resulting convex problem is proposed to be solved using FW, whose details are presented (including convergence).

Pros:
1. Though similar in spirit to SRW, the proposed formulation has few advantages: a) allows any cost, b) convex c) FW leads to scalable solver etc.
2. Overall, the paper is very well-written, with nice organization and sufficient details.

Cons:
1. Pre-fixed groups, more importantly, non-overlapping groups seems restrictive, especially because feature selection with overlapping groups is well-studied. (e.g., https://hal.inria.fr/inria-00628498/document , https://papers.nips.cc/paper/4275-efficient-methods-for-overlapping-group-lasso.pdf ) among others.

Major Comments:
1. Given SRW and other robust/min-max OT works, and multitude of feature-selection/group-lasso works, the novelty seems restricted. Even in terms of optimization, it seems a straight-forward application of FW. This seems to restrict the technical contribution.
2. In section 5.2, I am assuming for FROT, all layers were used as input; whereas for SRW, only few are used. Is this the case? If so, perhaps a case of FROT which uses exactly same input as SRW must be included for a fair comparison (along with the FROT with all layers). The authors do seem to agree that the improvement is more because of this skew in inputs. It is will nice to clarify this.
3. Why is that T is set in an adhoc manner? for example T=10 in synthetic and T=3 in real-world? why not fix or validate ? Also, convergence plots showing obj vs T as well as accuracy vs T might be insightful when included.
4. It may also be insightful to visually see some critical examples of image/pairs that highlight why FROT may work better than SRW etc. (more like fig5 in appendix)

---

> ### Author Response · Authors · 2020-11-20
> **Thank you for your feedback**
>
> Thank you for your valuable comments.
>
> **Given SRW and other robust/min-max OT works, and multitude of feature-selection/group-lasso works, the novelty seems restricted. Even in terms of optimization, it seems a straight-forward application of FW. This seems to restrict the technical contribution.**
>
> We agree that the formulation itself is similar to SRW. However, in our paper, our final goal is to solve the high-dimensional OT problems, where it has not been well studied. As clearly shown in the semantic correspondence experiments, SRW cannot directly be applied to such s high-dimensional data (d > 30,000).
>
> Moreover, we derive a new convex optimization problem Eq. (5). Thus, we believe the optimization researchers, who are in particular working for FW, can further improve and/or analyze the optimization of Eq. (5) by using newly developed FW methods.
>
> **In section 5.2, I am assuming for FROT, all layers were used as input; whereas for SRW, only few are used. Is this the case? If so, perhaps a case of FROT which uses exactly same input as SRW must be included for a fair comparison (along with the FROT with all layers). The authors do seem to agree that the improvement is more because of this skew in inputs. It is will nice to clarify this.**
>
> We have added the experiments in Table 1 (blue). As you can see, FROT compares favorably with SRW with the same set of layers. Therefore, we expect that SRW can achieve similar results of FROT by increasing the number of layers. However, at this point, the training of SRW for ultra high-dimensional data (i.e., all layers with d > 30,000) is infeasible.
>
> **Why is that T is set in an adhoc manner? for example T=10 in synthetic and T=3 in real-world? why not fix or validate ? Also, convergence plots showing obj vs T as well as accuracy vs T might be insightful when included.**
>
>  For the semantic correspondence tasks, since it converges with T=3, we simply set it to T = 3. For the revised manuscript, we have added T = 10 (See Table 1, red). As you can see, the performances of T = 3 and T = 10 are almost identical.
>
> **It may also be insightful to visually see some critical examples of image/pairs that highlight why FROT may work better than SRW etc. (more like fig5 in appendix)**
>
> We added more empirical results in Figure 7 (supplementary).

---

### Public Comment · ~Ievgen_Redko2 · 2020-11-17
**Clarification on Dhouib et al. ICML'20**

Great work! I have a question: when you say that our algorithm from Dhouib et al. ICML'20 may not converge for regularized optimal transport, do you mean the case where the transport matrix is initialized on the vertices?

---

> ### Author Response · Authors · 2020-11-19
> **Thank you for your feedback!**
>
> Thank you for your encouraging comments!
> We have not compared the proposed algorithm to your algorithm. Thus, we are not sure whether your algorithm really suffers the convergence issue in practice.
>
> According to your ICML paper,
>
> "Empirically, we observed that even the approximate solutions obtained by solving the entropy regularized formulation of the optimal transport problem ensure the convergence”
>
> your optimization technique also empirically works in practice. So, the empirical performance should not be that different from ours.
>
> Note that one of the advantages of the FW based algorithm proposed in our paper is that we can theoretically show the convergence of the algorithm using the regularized OT solvers.

---

### Decision · Program_Chairs · 2021-01-07
**Final Decision**

**Decision:**

Reject

**Comment:**

Motivated by (1) the problem of scaling up optimal transport to high-dimensional problems and (2) being able to tolerate noisy features, this paper introduces a new optimization problem that they call feature-robust optimal transport where they find a transport plan with discriminative features. They show that the min-max optimization problem admits a convex formulation and solve it using a Frank-Wolfe method. Finally they apply it to the layer selection problem and show that it achieves state-of-the-art performance for semantic correspondence datasets.

The reviews were mixed for this paper. The main negative, which was brought up in all the reviews, is the lack of novelty compared to earlier methods like SRW which already combine dimensionality reduction and optimal transport. The new method in this paper still does have value since it can scale up to larger dimensional problems. It would have been nice to have a wider range of experiments, which would present a more compelling case for its applicability. Another reviewer brought up a correctness issue, however it is not clear if this is actually a bug or merely a misunderstanding about how the pieces in the overall proof fit together. In any case, the reviewers pointed out various places where the writing could be improved.